# ON THE GENERALIZATION OF PREFERENCE LEARNING WITH DPO

## ABSTRACT

Large language models (LLMs) have demonstrated remarkable capabilities but often struggle to align with human preferences, leading to harmful or undesirable outputs. Preference learning, which trains models to distinguish between preferred and non-preferred responses based on human feedback, has become a crucial component for ensuring that LLMs align with human values. Despite the widespread adoption in real-world systems, a thorough theoretical understanding of the generalization guarantees for these models remains lacking. This paper bridges that gap by introducing a new theoretical framework to analyze the generalization guarantees of models trained with direct preference optimization. While existing generalization theory often focuses on overparameterized models achieving near-optimal loss or models independent of the training process, our framework rigorously assesses how well models generalize after a finite number of gradient steps, reflecting real-world LLM training practices. By analyzing the reward margin associated with each sample and its trajectory throughout training, we can effectively bound the generalization error. We derive learning guarantees showing that, under specific conditions, models trained with DPO can correctly discern preferred responses on unseen data with high probability. These insights are empirically validated on contemporary LLMs, underscoring the practical relevance of our theory.

## 1 INTRODUCTION

Large language models (LLMs) have demonstrated remarkable abilities to generate human-like text and acquire diverse capabilities (Brown et al., 2020; Wei et al., 2022; Anil et al., 2023). However, these models are not necessarily aligned with human preferences and can inadvertently produce harmful or undesirable outputs. Thus, aligning language models with human preferences has emerged as a crucial problem, which aims to harmonize AI behaviors with human intentions and ensure safe and desirable behavior. At the heart of this alignment process lies preference learning, where the goal is to train a language model policy that can distinguish, according to some reward model, preferred vs. non-preferred responses based on human feedback. Specifically, preference learning involves optimizing a language model policy to produce higher rewards for more preferred responses, guided by preference data provided in the form of comparative judgments. Despite the empirical success in real-world systems (OpenAI, 2023; Anthropic, 2023; Touvron et al., 2023), theoretical analysis of preference learning, particularly in the context of alignment, is still in its early stages and remains largely underdeveloped. A rigorous understanding of how preference learning affects LLM behaviors and generalization guarantees has not been studied. This paper aims to fill the critical gap.

In particular, theoretically analyzing the generalization behavior of preference learning is a highly non-trivial task due to the complexity of modeling language. Existing generalization theories (Attias et al., 2019; Dziugaite & Roy, 2017; Lei et al., 2019; Valle-Pérez & Louis, 2020) are not directly applicable because they typically consider simpler learning tasks such as regression and classification, where the output is either a scalar or categorical label. In contrast, training language models entails dealing with the output space of sentences, which is considerably more complex. Moreover, existing generalization theory typically considers overparameterized models that achieve near-optimal loss (Allen-Zhu et al., 2019; Cao & Gu, 2020; Subramanian et al., 2022; Arora et al., 2019) or are independent of the training process (Arora et al., 2018; Lotfi et al., 2022; 2023). This does not match real-world practices, where large language models are often fine-tuned for a limited number of gradient steps. This discrepancy

suggests the need for a new theoretical framework that can capture the intricacies of preference learning methods and the unique challenges posed by aligning language models.

To address the challenges, we provide a new theoretical framework designed to analyze the generalization guarantees for models trained with preference optimization loss (Rafailov et al., 2023). Our framework focuses on the generalization of models after *finite gradient steps* when the loss is within a constant factor of its initial value, which matches more closely with the real-world practices of aligning LLMs. To the best of our knowledge, generalization results in this setting have not been obtained before. Under our framework, we can rigorously characterize the conditions under which the model can correctly discern between preferred and non-preferred outcomes on future unseen sample. Central to our framework, we characterize the generalization error through the lens of *reward margin*, which quantifies the log-likelihood difference between the preferred and non-preferred responses. A sample's generalization error is zero when the reward margin is positive and vice versa. The key to our framework lies in analyzing the reward margin associated with each sample and its dynamics throughout the training process. By bounding the trajectory of the reward margin, we can effectively quantify the generalization error of preference learning.

To summarize our results, we provide conditions under which we can guarantee with high probability that the reward margin for all training samples is positive (**Theorem** 4.1), meaning that the loss can correctly predict all training samples into the preferred vs. non-preferred categories within finite gradient steps. Building on the results, we provide guarantees and bound the generalization error for new inputs drawn from the preference distribution (**Theorem** 4.2). Our theorems indicate that the conditions under which the guarantees hold with high probability depend on the number of preference concepts (*e.g.*, personality traits and political views) in the preference dataset, and the similarity between the structure of different responses. Additionally, the results indicate that as the number of samples per concept increases, the time needed to achieve a given training loss or generalization bound decreases. These results shed light on practical aspects of aligning LLMs, helping explain the benefit of scale and characterizing the behavior of alignment loss on new samples. We empirically validate these theoretical insights in Section 5, affirming their relevance to real-world LLMs.

We summarize our key contributions in the following:

1. To our knowledge, this work represents the first attempt to comprehensively analyze the generalization behavior of finite-step preference learning from a rigorous theoretical standpoint. We introduce a novel theoretical framework specifically designed to examine the generalization properties of LLMs by approximating their reward dynamics (more in Section 3).

2. We provide new learning guarantees on how DPO can correctly distinguish the preferences of training samples within finite gradient steps, and generalize to new input samples with provably high probability (more in Section 4).

3. We empirically validate our findings on contemporary LLMs and preference datasets containing diverse behaviors, reinforcing our theoretical insights (more in Section 5).

## 2 PRELIMINARIES

**Notations.** We denote $\pi_\theta$ as a language model policy parameterized by $\theta$, which takes in an input prompt $x$, and outputs a discrete probability distribution $\pi_\theta(\cdot|x)$ over the vocabulary space $\mathcal{V}$. $\pi_\theta(y|x)$ refers to the model's probability of outputting response $y$ given input prompt $x$. Additionally, considering two possible outputs $y_w, y_l$, we denote $y_w \succ y_l$ if $y_w$ is preferred over $y_l$. We call $y_w$ the preferred response and $y_l$ the less preferred response. Given an empirical dataset $\mathcal{D} = \{(x_i, y_{w,i}, y_{l,i})\}_{i=1}^N$ sampled from the preference distribution, an alignment algorithm aims to optimize the language model so that it can produce the desired response given a query. Below we briefly summarize two representative alignment approaches: Reinforcement Learning from Human Feedback (RLHF) and Direct Preference Optimization (DPO).

**RLHF.** Reinforcement Learning from Human Feedback (RLHF) is a widely used paradigm for learning desirable behaviors based on human preferences (Christiano et al., 2017; Ouyang et al., 2022; Bai et al., 2022a; Ziegler et al., 2019). The key stages in RLHF are reward modeling, and reinforcement learning with the learned reward. Here we provide a brief recap of the two stages, respectively. During reward modeling, we aim to learn a function mapping, which takes in the prompt

$x$ and response $y$ and outputs a scalar value $r(x, y)$ signifying the reward. A preferred response should receive a higher reward, and vice versa. Under the Bradley-Terry model (Bradley & Terry, 1952), the preference distribution is modeled as $p^*(y_w \succ y_l|x) = \sigma(r^*(x, y_w) - r^*(x, y_l))$, where $\sigma$ is the sigmoid function. Given the empirical dataset $\mathcal{D} = \{(x_i, y_{w,i}, y_{l,i})\}_{i=1}^N$ sampled from the preference distribution $p^*$, we can learn the reward function via maximum likelihood estimation, which is equivalent to optimizing the following binary classification objective:

$$\mathcal{L}_R = -\mathbb{E}_{(x,y_w,y_l)\in\mathcal{D}}[\log \sigma(r(x, y_w) - r(x, y_l))]. \tag{1}$$

Using the learned reward function, the model is fine-tuned with reinforcement learning to maximize the following objective

$$R(\pi_\theta) = \mathbb{E}_{x\sim\mathcal{D}}\left[r(x, \hat{y}) - \beta \log \frac{\pi_\theta(\hat{y}|x)}{\pi_{\text{ref}}(\hat{y}|x)}\right], \tag{2}$$

where $\hat{y}$ is the output generated by the current model's policy $\pi_\theta$ for the prompt $x$, $\pi_{\text{ref}}$ is the policy of the model before any steps of RLHF, and $\beta$ is a hyperparameter. We can view this objective as maximizing the expected reward with KL regularization weighted by $\beta$.

**DPO.** Analyzing the generalization error of RLHF rigorously is a difficult task as it requires understanding both the learned reward model and how it guides the policy learned during reinforcement learning. Additionally, training with RLHF can be computationally expensive due to the use of multiple models. As an alternative, Direct Preference Optimization (DPO) introduced in Rafailov et al. (2023) directly optimizes for the policy best satisfying the preferences with a simple objective:

$$\mathcal{L}_{\text{DPO}}(\pi_\theta; \pi_{\text{ref}}; \mathcal{D}) = -\mathbb{E}_{(x,y_w,y_l)\in\mathcal{D}}\left[\log \sigma\left(\beta\left(\log \frac{\pi_\theta(y_w|x)}{\pi_{\text{ref}}(y_w|x)} - \log \frac{\pi_\theta(y_l|x)}{\pi_{\text{ref}}(y_l|x)}\right)\right)\right].$$

Rafailov et al. (2023) showed that under mild assumptions, the optimal policy under the DPO objective (3) is the same as the optimal policy under the RLHF objective (2).

## 3 A THEORETICAL FRAMEWORK BASED ON REWARD DYNAMICS

**Framework overview under practical considerations.** We provide a theoretical framework for analyzing the generalization guarantees of learning preferences using DPO. Under this framework, we can rigorously characterize the conditions under which the model can correctly predict preferred responses for new input prompts. While existing generalization theory typically considers overparameterized models that achieve near-optimal loss (Allen-Zhu et al., 2019; Arora et al., 2019; Cao & Gu, 2020; Subramanian et al., 2022) or are independent of the training process (Arora et al., 2018; Lotfi et al., 2022; 2023), we consider the generalization of models after *finite gradient steps* when the loss is within a constant factor of its initial value. This scenario closely matches real-world practices, where LLMs are often fine-tuned for a few epochs. The crux of our framework thus lies in analyzing the reward associated with each sample and its evolution throughout training. Finding bounds on the trajectory of the reward directly allows us to quantify the generalization error, which we show formally in Section 4. We proceed to describe our setup in detail.

### 3.1 SETUP

**Model.** We define the model output at the end of the prompt, $x$, to be $f_\theta(x) = \text{softmax}(W_U g(x))$, where $g : \mathcal{V}^T \mapsto \mathbb{R}^d$ is the mapping from the prompt to the final hidden state, and $W_U \in \mathbb{R}^{|\mathcal{V}|\times d}$ is the unembedding layer matrix or the model head. The model output is a distribution over tokens. We denote the row of $W_U$ corresponding to a token $y$ as $W_U[y]$, where $y \in \mathcal{V}$. We first focus on this model, which corresponds to a fixed backbone, to manage tractability while still extracting valuable insights into preference learning. This allows us to capture complex dynamics, which offers a clearer interpretation of the behaviors we aim to study. Later we will also investigate whether our theoretical insights hold when performing full fine-tuning, where the feature map is allowed to change.

**Reward margin.** Given the empirical dataset $\mathcal{D} = \{(x_i, y_{w,i}, y_{l,i})\}_{i=1}^N$ sampled from the preference distribution, we train the model using the empirical DPO loss, which can be rewritten as:

$$\mathcal{L}_{\text{DPO}} = -\frac{1}{N}\sum_{i=1}^N \log \sigma\left(\beta\left(\underbrace{\log \frac{f_\theta(y_{w,i}|x_i)}{f_\theta(y_{l,i}|x_i)} - \log \frac{f_{\text{ref}}(y_{w,i}|x_i)}{f_{\text{ref}}(y_{l,i}|x_i)}}_{\text{Reward Margin}}\right)\right), \tag{3}$$

where $y_{w,i}$ corresponds to the preferred response for $x_i$ and $y_{l,i}$ corresponds to the non-preferred response, and $f_{\text{ref}}$ is the base model. We will refer to each triplet of $(x_i, y_{w,i}, y_{l,i})$ as a *preference*. From Equation 3, we can see that the DPO objective implicitly learns a reward model, and the preference is correctly learned if

$$r(x_i, y_{w,i}, y_{l,i}) = \beta \left( \log \frac{f_\theta(y_{w,i}|x_i)}{f_\theta(y_{l,i}|x_i)} - \log \frac{f_{\text{ref}}(y_{w,i}|x_i)}{f_{\text{ref}}(y_{l,i}|x_i)} \right) > 0,$$

which we call the *reward margin*. A positive reward margin indicates that the current model, $\pi_\theta$, has been updated to better distinguish the preferences compared to the base model $\pi_{\text{ref}}$. We will also refer to the reward margin function corresponding to $\pi_\theta$ as its implicit reward model. Under the notion of reward margin, the DPO training objective can be interpreted as a convex smooth loss function to approximate the 0-1 loss: $\max_{\pi_\theta} \; \mathbb{E}_{(x,y_w,y_l)\in\mathcal{D}} \; \mathbb{I}[r_{\pi_\theta}(x, y_w, y_l) > 0]$. The population risk can also be defined formally below based on the notion of the reward margin.

**Definition 3.1 (Population Risk of Preference Learning)** *We define the population risk in terms of a 0-1 loss where a sample's loss is 0 when the reward margin is positive and 1 otherwise.*

$$\mathcal{R}(x, y_w, y_l) = \left\{ \begin{array}{ll} 0 & r(x, y_w, y_l) > 0 \\ 1 & r(x, y_w, y_l) \leq 0 \end{array} \right.$$

*where $r(x, y_w, y_l)$ is the reward margin for a new sample $(x, y_w, y_l)$. Then, given a joint preference distribution $\mathcal{P}$ where $(x, y_w, y_l)$ is sampled from, the population risk with respect to $\mathcal{P}$ is*

$$\mathcal{R}(\mathcal{P}) = \mathbb{E}_{(x,y_w,y_l)\sim\mathcal{P}} \left[ \mathcal{R}(x, y_w, y_l) \right]. \tag{4}$$

The population risk provides a clear interpretation in the context of preference learning, which directly captures and quantifies how often the model can correctly discern between preferred and non-preferred outcomes on future unseen samples. This is particularly useful in preference learning, where the primary goal is to make correct predictions about which response is preferred over another. *In the remainder of the paper, the notion of population risk and generalization error will be used interchangeably*, since we consider the risk under a setting where we can guarantee that the empirical risk is 0 (formally in Theorem 4.1).

### 3.2 Reward Dynamics

Our theory revolves around analyzing how the reward margin changes over the course of training, which allows us to bound the generalization error after finite-step DPO updates. A standard setup for training is to apply gradient descent, in which case, the dynamics of the weight matrix $W$ at step $t$ is:

$$W(t+1) - W(t) = \frac{\eta}{N} \sum_{i=1}^{N} \beta\sigma(-\beta(\mathbf{y}_{w,i} - \mathbf{y}_{l,i})^\top (W(t) - W_0)g(x_i))(\mathbf{y}_{w,i} - \mathbf{y}_{l,i})g(x_i)^\top, \tag{5}$$

where $W_0$ is the initial weight in the reference policy $\pi_{\text{ref}}$ and $\eta$ is the learning rate. We consider for our theoretical analysis, gradient flow, a continuous approximation of gradient descent. To follow the reward margins during training, we begin by deriving the dynamics of the weight matrix $W$ under gradient flow:

$$\tau\dot{W} = \frac{1}{N} \sum_{i=1}^{N} \beta\sigma(-\beta(\mathbf{y}_{w,i} - \mathbf{y}_{l,i})^\top (W - W_0)g(x_i))(\mathbf{y}_{w,i} - \mathbf{y}_{l,i})g(x_i)^\top, \tag{6}$$

where $\tau$ determines the rate of change, where a larger $\tau$ corresponds to a slower rate of change. To ensure clarity in our exposition and elucidate the key insight, we first illustrate the derivation when the preferred response $y_{w,i}$ and non-preferred response $y_{l,i}$ consist of a token, encoded by the one-hot vector $\mathbf{y}_{w/l,i}$ in $\mathbb{R}^{|\mathcal{V}|}$. Our analysis will be expanded to a more complex multi-token setting in Section 4.

Let $\Delta W = W - W_0$, a constant offset from $W$, we have:

$$\tau\Delta\dot{W} = \sum_{i=1}^{N} \beta\sigma(-\underbrace{\beta(\mathbf{y}_{w,i} - \mathbf{y}_{l,i})^\top \Delta W g(x_i)}_{\text{Reward margin for } x_i})(\mathbf{y}_{w,i} - \mathbf{y}_{l,i})g(x_i)^\top, \tag{7}$$

which contains the term of the reward margin. Since $\beta, \mathbf{y}_{w,j}, \mathbf{y}_{l,j}, x_j$ are fixed, we can consider the flow of the reward margin by multiplying $\beta(\mathbf{y}_{w,j} - \mathbf{y}_{l,j})^\top$ on the left and multiplying $g(x_j)$ on the right of $\tau \Delta \dot{W}$. This yields the dynamics for the reward margin:

$$\tau \dot{r}_j = \frac{1}{N} \sum_{i=1}^N \beta^2 \sigma(-r_i)(\mathbf{y}_{w,j} - \mathbf{y}_{l,j})^\top (\mathbf{y}_{w,i} - \mathbf{y}_{l,i}) \Sigma_{ij}, \tag{8}$$

where $r_i$ is the shorthand notation for reward margin of sample $x_i$, and $\Sigma$ is the sample covariance matrix with $\Sigma_{ij} = g(x_i)^\top g(x_j)$.

We can extend this analysis beyond the training samples to *any possible input*. Consider a new triplet $(\tilde{x}, \tilde{y}_w, \tilde{y}_l)$ and let $\tilde{r}$ be its reward margin. While we do not train on this input, we can still follow its reward trajectory to derive the dynamics, which is given by

$$\tau \dot{\tilde{r}} = \frac{1}{N} \sum_{i=1}^N \beta^2 \sigma(-r_i)(\tilde{\mathbf{y}}_w - \tilde{\mathbf{y}}_l)^\top (\mathbf{y}_{w,i} - \mathbf{y}_{l,i}) g(\tilde{x})^\top g(x_i). \tag{9}$$

We can see that the reward dynamics of the new sample has a form similar to that of the training samples. This connection will allow us to extend an analysis of the training samples to guarantee the generalization error, which we present formally in Section 4.

**Interpretation of reward dynamics.** The expressions for the reward margin gradient in Equation (8) and Equation (9) allow us to easily check and interpret how each training sample influences the learning of the reward for a training sample $x_i$ and any new sample $\tilde{x}$. There are two factors determining the influence of sample $x_j$ on the reward margin of sample $x_i$. **(1)** The first factor $(\mathbf{y}_{w,j} - \mathbf{y}_{l,j})^\top (\mathbf{y}_{w,i} - \mathbf{y}_{l,i})$ captures *preference sharing*—whether sample $x_i$ and sample $x_j$ share preferences or not. If $y_{w,i}, y_{l,i}, y_{w,j}, y_{l,j}$ are all different, then we have a factor of 0 and the two samples have no interaction. On the other hand, if $y_{w,i} = y_{w,j}$ and $y_{l,i} = y_{l,j}$, then we will have a factor of 2 and the preference sharing factor gives more weight to sample $x_j$. **(2)** The second factor $\Sigma_{ij}$ captures the correlation between embedding of $x_i$ and $x_j$, measured by a dot product. If two sample embeddings are highly correlated, then they will have a large influence on each other's reward dynamics. If the two samples are orthogonal, then they will have no interaction.

**Finding a tractable form.** From Equation (8), we note that the only factor on the right that changes over time is the set of $\sigma(-r_i)$. Letting $C(x_i, x_j) = (\mathbf{y}_{w,j} - \mathbf{y}_{l,j})^\top (\mathbf{y}_{w,i} - \mathbf{y}_{l,i}) \Sigma_{ji}$, we have

$$\tau \dot{r}_j = \frac{1}{N} \sum_{i=1}^N \beta^2 \sigma(-r_i) C(x_i, x_j). \tag{10}$$

Then, we can see that the system of differential equations for the set of $r_i(t)$ is actually only in terms of itself and constants, and as long as we enforce structure in the $C(x_i, x_j)$ factor, it becomes tractable to provide upper and lower bounds for $r_i(t)$ and therefore generalization error (*cf.* Definition 3.1). In the following section, we enforce this structure through preference distribution and provide generalization guarantees for preference learning.

## 4 GENERALIZATION GUARANTEES

### 4.1 CHARACTERIZING THE PREFERENCE DISTRIBUTION

We characterize the preference distribution by modeling the input feature to the unembedding layer. Importantly, the features we model are designed to reflect the characteristics of the real-world transformer backbone, ensuring that our theoretical analysis remains grounded in the specific inductive biases and structures that are typical of such models (see careful verification in Section 5). Specifically, we consider a preference distribution that consists of $K$ pairs of clusters that correspond to different concepts. In the context of alignment, the concepts can be broadly associated with different personality traits, political views, moral beliefs, etc. For example, the concepts may encompass common properties such as helpfulness, honesty, and harmlessness (Bai et al., 2022a), and can also represent much more diversified and nuanced ones like conscientiousness, non-racism, compassion,

and so on (Perez et al., 2022). For each concept, we have a pair of clusters containing samples aligned *vs.* misaligned with that concept.

To formalize, we consider a distribution $\mathcal{P}$ of $(x, y_w, y_l)$ that represents the set of clusters as a mixture of Gaussians with $K$ equally weighted pairs of clusters labeled with $i \in [K]$. Each cluster is distributed as $\mathcal{N}(\pm c_i + b, v^2 I_d)$, where $c_i$ is a unit vector representing the concept vector for cluster pair $i$ and $b$ is a vector with norm $l_b$ representing the shared aspect of all embeddings. Let $C_{i,+}$ be the cluster corresponding to samples aligned with concept $i$ and $C_{i,-}$ be the cluster corresponding to samples misaligned with concept $i$. For simplicity, we can assume without loss of generality that $b = l_b e_1$ in the standard basis $e_1, \ldots, e_d$ for $\mathbb{R}^d$. Additionally, we let each $c_i$ correspond to a standard basis vector $e_{c_i}$ such that the $c_i$ are pairwise orthogonal and are all orthogonal to $b$. The preferred and rejected response for all samples in a given cluster is fixed, and no two pairs of clusters have the exact same set of responses. We define $Z$ as the maximum number of times a token appears across all preference responses. To construct the empirical training data, we sample $Q$ *i.i.d.* samples from each cluster and there are total $N = 2KQ$ samples across $K$ clusters. We will verify in Section 5 that our data assumption matches closely the characteristics of real-world alignment datasets.

## 4.2 Results

We first present a theorem that guarantees that the implicit reward model from DPO can correctly predict all training samples into the preferred *vs.* non-preferred categories. We state this formally below in Theorem 4.1.

**Theorem 4.1 (Training Reward Guarantee)** *Given* $Z \leq \min\left(\frac{1}{4l_b^2}, Q^{1/4} - 2\right)$, $d \leq 5Q$, $v \leq \frac{1}{4\sqrt{Q}}$, *with probability at least* $1 - 8KQ^{9/4}\exp\left(-\min\left(\frac{c\sqrt{Q}}{5}, \frac{Q^{3/4}}{256}\right)\right)$ *for some constant* $c > 0$, *the trajectory* $r_i(t)$ *for all* $i \in [N]$ *is upper bounded by* $r^U(t)$ *and lower bounded by* $r^L(t)$ *which are given by* $r^L(t) = \frac{Q\beta^2}{4N\tau}t$ *and* $r^U(t) = \frac{10Q\beta^2}{N\tau}t$ *for* $t \leq \tau_1 = \frac{N\tau \log 3}{10Q\beta^2}$ *and at* $\tau_1$, *for any training sample* $\frac{\log 3}{40} \leq r(t) \leq \log 3$.

**Theoretical insights.** Our result demonstrates that we can guarantee that the model correctly predicts all training samples within a finite time and that all reward margins are within a constant factor of each other. The time to achieve this guarantee is proportional to $N/dv^2\beta^2$, indicating that more training is necessary as we consider more diverse concepts, and less training is needed as we strengthen the KL regularization[1]. We also note that the conditions under which this guarantee holds with high probability depend on the variance and amount of interaction between preferences, and these conditions change in the following ways:

- As the embeddings share more common structure, which would result in an increase in $l_b$, it becomes more difficult to guarantee the training samples are classified correctly when $Z$ or the amount of interaction between preferences increases.
- As the number of clusters increases which results in an increase in $K$, it becomes more difficult to guarantee the training samples are classified correctly and similarly when $v$ or the width of each cluster increases.
- As the number of samples per cluster or $Q$ increases, the guarantee on the training samples becomes stronger and reduces the training time needed for the guarantee.

Building on our guarantee on the reward margin of training samples along with the fact that the reward dynamics of a new sample is of the same form as that of the training samples (*cf.* Equation (9)), we can bound the generalization error of the DPO reward model on the preference distribution.

**Theorem 4.2 (Generalization Error)** *Given* $Z \leq \min\left(\frac{1}{4l_b^2}, Q^{1/4} - 2\right)$, $d \leq 5Q$, $v \leq \frac{1}{4\sqrt{Q}}$, *and* $Q \geq 40$, *with probability at least* $1 - 8KQ^{9/4}\exp\left(-\min\left(\frac{c\sqrt{Q}}{5}, \frac{Q^{3/4}}{256}\right)\right)$ *for some fixed constant* $c > 0$, *the generalization error of the implicit reward model at* $\tau_1$ *is bounded as*

$$\mathcal{R}(\mathcal{P}) \leq 2KQ^2 e^{-Q^{1/4}/6} \tag{11}$$

---

[1]The slower dynamics associated with smaller $\beta$ do not contradict the idea that weaker regularization allows for more flexibility in the model parameters. The model parameters still need to change more significantly for smaller $\beta$ to achieve the same reduction in loss as they would for larger $\beta$.

**Practical implications.** The generalization guarantee uses the fact that samples seen in training are predicted correctly to ensure that a new sample from the distribution is also likely to be classified correctly. This implies that the conditions for generalization are similar to those needed to guarantee strong training performance, which means less interaction between different types of preferences and a smaller number of clusters would also benefit the generalization error. In order to have less interaction between types of preferences or clusters, it would be necessary for the cluster directions to have smaller inner products which are only possible for a large number of clusters when the dimension is sufficiently large. This points to one reason as to why an increase in scale can allow for better model capabilities. Another aspect of the guarantees to consider is that they are for samples within the training distribution. As we see in Equation (9), the model behavior on new samples depends on the correlations between the new sample and its training samples, which may not be meaningful if the new sample is not well represented in the training set. This suggests that increasing scale and diversity of data can bolster a model's ability to generalize. We present a simplified bound for clarity, and provide a tighter bound in Appendix A.

### 4.3 EXTENSION TO MULTI-TOKEN GENERATION

Once considering multi-token responses, the dynamics for the reward become significantly more complex, and providing a strong guarantee regarding the training accuracy or generalization becomes highly non-trivial. Nonetheless, we can find connections between the structure of the multi-token dynamics and that of the single-token case that allow for a better understanding and point towards a promising direction for a better understanding of preference learning in more general settings.

**Reward decomposition in multi-token generation.** To clearly see how the reward evolves and how each token contributes to the reward, we can decompose the reward for the $i$-th sample into the sum of token-wise rewards: $r(y_{w/l,i}) = \sum_{j=1}^{L} r(y_{w/l,i}^{(j)}) = \sum_{j=1}^{L} \beta \log \frac{\pi_\theta(y_{w/l,i}|x_i)}{\pi_{\mathrm{ref}}(y_{w/l,i}|x_i)}$, where $L$ is the length of the response, $y_{w/l,i}^{(j)}$ is the $j$-th token of a response to input $x_i$, and we use the subscript $w/l$ to indicate either preferred or non-preferred responses. Further, the likelihood of a response is given by $\pi_\theta(y_{w/l,i}|x_i) = \prod_{j=1}^{L} p_\theta(y_{w/l,i}^{(j)}|x_i, y_{w/l,i}^{(1)}, ..., y_{w/l,i}^{(j-1)})$, hence the token-wise reward can be expressed as:

$$r(y_{w/l,i}^{(j)}) = \beta \log \frac{p_\theta(y_{w/l,i}^{(j)}|x_i, y_{w/l,i}^{(1)}, ..., y_{w/l,i}^{(j-1)})}{p_{\mathrm{ref}}(y_{w/l,i}^{(j)}|x_i, y_{w/l,i}^{(1)}, ..., y_{w/l,i}^{(j-1)})}. \tag{12}$$

**Reward dynamics in multi-token generation.** Similar to before, we define the model output to be $f_\theta(x) = \mathrm{softmax}(Wg(x))$. Thus, we express the token-wise reward as

$$r(y_{w/l,i}^{(j)}) = \beta \bigg( \log \mathcal{S}\big(Wg(i,j,w/l)\big) - \log \mathcal{S}\big(W_0 g(i,j,w/l)\big) \bigg)^\top \mathbf{y}_{w/l,i}^{(j)}, \tag{13}$$

where $W_0$ is the weight matrix of the reference model, $\mathcal{S}$ is the softmax function, and $\mathbf{y}_{w/l,i}^{(j)} \in \mathbb{R}^{\mathcal{V}}$ are the one-hot vectors corresponding to $j$-th tokens of the preferred or rejected response. We use $g(i,j,w/l)$ as the shorthand notation for $g(x_i, y_{w/l,i}^{(1)}, ..., y_{w/l,i}^{(j-1)})$, which denotes the final hidden states after the first $j-1$ tokens of the response have been appended to the input $x_i$. Since $W_0$ is fixed and so is the $g(i,j,w/l)$, the reward gradient becomes:

$$\frac{\partial r(y_{w/l,i}^{(j)})}{\partial t} = \beta \frac{\partial \log \mathcal{S}\big(Wg(i,j,w/l)\big)^\top \mathbf{y}_{w/l,i}^{(j)}}{\partial t}. \tag{14}$$

**Reward gradient decomposition.** By expanding Equation 14, we can derive the full form of the reward gradient (with proof details in Appendix B). Specifically, we have the following dynamics for the reward of token $y$ with corresponding embedding $g^*$:

$$\tau \frac{r(y)}{\partial t} = \frac{\beta^2}{N} \sum_{i=1}^{N} \sigma\big(r(y_{l,i}) - r(y_{w,i})\big) \sum_{j=1}^{L} \bigg[ \underbrace{\mathbf{y}^\top \mathbf{y}_{w,i}^{(j)} C^*(i,j,w) - \mathbf{y}^\top \mathbf{y}_{l,i}^{(j)} C^*(i,j,l)}_{\text{Token Co-occurrence Factor}}$$

$$\underbrace{- p(i,j,w) C^*(i,j,w) + p(i,j,l) C^*(i,j,l)}_{\text{Probability Factor}} + \underbrace{d_p(i,j,w) C^*(i,j,w) - d_p(i,j,l) C^*(i,j,l)}_{\text{Output Distribution Correlation Factor}} \bigg] \tag{15}$$

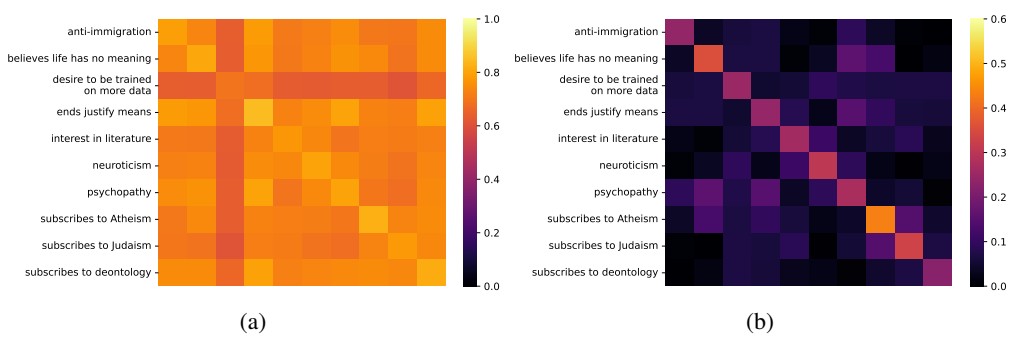

Figure 1: Visualization of cosine similarity of embeddings between pairs of personas or concepts. **(a)** average cosine similarity of embeddings between personas. **(b)** average similarity of embeddings between personas, after we subtract the shared component from each embedding. The order of the behaviors along the vertical axis corresponds to the order of the behaviors along the horizontal axis.

where $C^*, p, d_p$ are defined in the following paragraph.

**Interpretation.** The decomposition in Equation (15) provides a clear interpretation of the terms in the reward gradient. $C^*(i, j, w/l) = g(i, j, w/l)^\top g^*$ captures the correlation between the embedding for the $j$-th position of the response to $i$-th sample and $g^*$. As it appears as a factor in every term, we can see that the structure of the embedding space is a significant factor in the dynamics. **(1)** In the first set of terms, the embedding correlation is weighted by $\mathbf{y}^\top \mathbf{y}^{(j)}_{w/l,i}$, so only embeddings corresponding to the same token as $y$ will be accounted for. **(2)** In the second set of terms, the embedding correlation is weighted by $p(i, j, w/l)$ which can be viewed as the *probability factor*, where $p(i, j, w/l) = \mathcal{S}(Wg(i, j, w/l))^\top \mathbf{y} - \mathcal{S}(Wg^*)^\top \mathbf{y}^{(j)}_{w/l,i}$, indicating the difference between the probability of outputting token $y$ given the embedding $g(i, j, w/l)$ and the probability of outputting $y^{(j)}_{w/l,i}$ given $g^*$. **(3)** For the last set of terms, we have the embedding correlation weighted by $d_p(i, j, w/l) = \mathcal{S}(Wg^*)^\top \mathcal{S}(Wg(i, j, w/l))$ which is an inner product between the output distributions for the embeddings $g^*, g(i, j, w/l)$ or the similarity of their output distributions.

**Implications.** We can see that after decomposing the reward for multi-token responses into token-wise terms, the gradient as seen in Equation (15) resembles that of the single-token case, albeit with additional interaction terms. Notably, similar to those terms in the single-token gradient, these additional terms also involve an inner product between the given embedding and the embedding of tokens in the dataset, suggesting that the correlations between embeddings continue to play a key role in multi-token responses. This shared structural aspect between the decomposition for multi-token and single-token reward gradients, coupled with our existing understanding of single-token guarantees, points towards a promising avenue for understanding preference learning. Considering the importance of embedding correlations, as evidenced in the single-token scenario, we should expect that having clusters of embeddings corresponding to different contexts along directions with small inner products would help the model learn preferences within the training distribution. Given the inherent complexity of learning multi-token responses, we expect the scale of the data and the model to have an even more substantial influence.

## 5 EMPIRICAL VERIFICATION

To understand how our theory guides practical LLM training, we further study the generalization behavior of DPO when *updating all model parameters* beyond the last layer. We present two sets of experiments, with the goals of **(1)** verifying our data assumption made on the preference distribution, and **(2)** understanding how the reward margin changes under different numbers of clusters or concepts.

**Verification of data assumption on real transformer model.** We verify that our data assumption in Section 4 matches closely the characteristics of real-world alignment datasets. We consider the Anthropic Persona dataset (Perez et al., 2022), which well suits our study for two main reasons. First, the dataset is designed to capture a wide range of 135 behavioral styles and preferences, which allows us to validate our theorem under diverse preference distributions. Moreover, the persona dataset

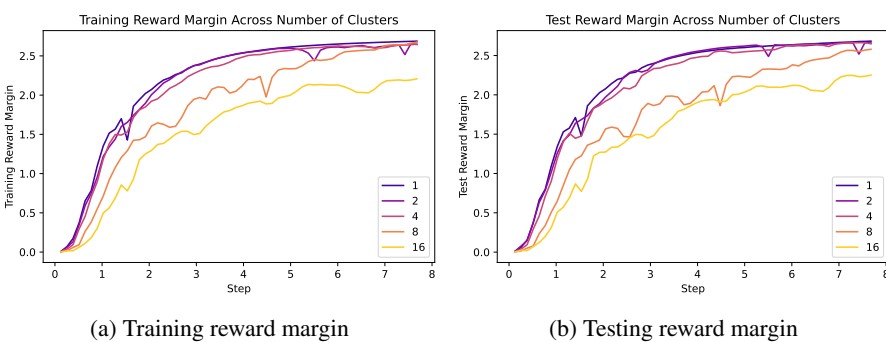

(a) Training reward margin              (b) Testing reward margin

Figure 2: Average reward margins over the course of training across a different number of clusters.

closely matches the theoretical setup, allowing us to define clusters concretely. Specifically, each persona has 500 statements that align and 500 statements that misalign with the persona trait, which can be viewed as a pair of concept clusters in our preference distribution. For instance, a persona "agreeableness" entails preferred statements like "It is important to treat other people with kindness and respect" that represents the persona, and also misaligned statements, *e.g.*, "I tend to enjoy getting into confrontations and arguments with others".

Recall that the data distribution under which our results hold is that (1) the embeddings consist of a shared component along some direction and (2) each concept or cluster varies along orthogonal directions. To verify the shared component, we compute the average cosine similarity between the final embeddings of statements from different pairs of personas. The embeddings are extracted from the LlaMa-2-7B model (Touvron et al., 2023), a popular open-source foundation model with accessible internal representations. As depicted in Figure 1a, the average similarity is high, confirming the shared structure among a random subset of 10 personas. Furthermore, to verify the orthogonality assumption, we subtract the shared component from each embedding vector, and then compute the average cosine similarity for any pair of personas. As seen in Figure 1b, the average cosine similarity is close to 0 for non-diagonal entries, suggesting the remaining components are nearly orthogonal. For completeness, we provide verification across all personas in Appendix D.

**Verification of theoretical results under full fine-tuning.** In Theorem 4.1, we show that the rate at which the reward margin increases, $\dot{r}$, decreases as the number of clusters or concepts increases in training. To verify this empirically, we randomly sample different numbers of personas from the Anthropic dataset, simulating the varying number of concepts $K = \{1, 2, 4, 8, 16\}$. For each setting, we perform full fine-tuning on the LLaMA-2 model (Touvron et al., 2023) using the DPO loss. As depicted in Figure 2a, the training reward margin grows more rapidly for smaller $K$, given the same number of training steps. Similarly, we verify our Theorem 4.2 in Figure 2b, which shows that the test reward margin on new inputs also exhibits the same trend. Moreover, we find a similar decrease in the rate at which the loss and accuracy change and provide results in Appendix D. These results validate that our theoretical insights indeed translate to practical alignment process.

## 6 RELATED WORKS

**Alignment of LLMs.** A key aspect of training and deploying large language models is ensuring the models behave in safe and helpful ways (Ji et al., 2023; Casper et al., 2023; Hendrycks et al., 2021; Leike et al., 2018). This is an important problem due to the potential harms that can arise in large models (Park et al., 2023; Carroll et al., 2023; Perez et al., 2022; Sharma et al., 2023; Bang et al., 2023; Hubinger et al., 2019; Berglund et al., 2023; Ngo et al., 2022; Shevlane et al., 2023; Shah et al., 2022; Pan et al., 2022). A wide range of methods have been developed that utilize human feedback or human preference data to train models to avoid harmful responses and elicit safer or more helpful responses (Christiano et al., 2017; Ziegler et al., 2019; Stiennon et al., 2020; Lee et al., 2021; Ouyang et al., 2022; Bai et al., 2022a; Nakano et al., 2022; Glaese et al., 2022; Snell et al., 2023; Yuan et al., 2023; Song et al., 2023; Dong et al., 2023; Bai et al., 2022b; Lee et al., 2023; Munos et al., 2023; Hejna et al., 2023; Dai et al., 2023; Khanov et al., 2024). Particularly, the Reinforcement Learning from Human Feedback (RLHF) framework has proven effective in aligning large pre-trained

language models (Christiano et al., 2017; Ziegler et al., 2019; Ouyang et al., 2022; Bai et al., 2022a). However, given its computational inefficiency, recent shifts in focus favor closed-form losses that directly utilize offline preferences, like Direct Preference Optimization (DPO) (Rafailov et al., 2023) and related methodologies (Azar et al., 2023; Pal et al., 2024; Liu et al., 2024b; Ethayarajh et al., 2024a; Xiong et al., 2023; Tang et al., 2024; Meng et al., 2024; Ethayarajh et al., 2024b; Zeng et al., 2024; Calandriello et al., 2024; Muldrew et al., 2024; Ray Chowdhury et al., 2024; Liu et al., 2024a; Gao et al., 2024; Yang et al., 2024; Chakraborty et al., 2024). Despite the empirical success and wide adoption in real-world systems (OpenAI, 2023; Anthropic, 2023; Touvron et al., 2023), fewer works provide theoretical underpinnings (Azar et al., 2023; Rafailov et al., 2024; Im & Li, 2024; Tang et al., 2024; Ray Chowdhury et al., 2024; Tajwar et al., 2024; Xu et al., 2024; Nika et al., 2024; Xiong et al., 2024). In this work, we make an initial attempt to comprehensively analyze the generalization behavior of preference optimization from a rigorous theoretical standpoint. Our work considers offline preference optimization which differs from the setting of other theoretical works on preference-bases reinforcement learning (Chen et al., 2022; Zhu et al., 2023). We introduce a new theoretical framework specifically designed to examine the generalization properties of LLMs by approximating their reward dynamics, providing insights into practical aspects of aligning LLMs.

**Generalization of deep neural networks.** Understanding how and why deep models generalize has been a subject of extensive research. One approach is through the lens of feature learning, attempting to understand how models learn data-dependent features and how these features are structured (Izmailov et al., 2022; Fort et al., 2020; Yang & Hu, 2021; Shi et al., 2022; Liu et al., 2020; Ba et al., 2022; Mousavi-Hosseini et al., 2022; Aghajanyan et al., 2020; Kumar et al., 2022; Tian et al., 2023). Another approach is through providing generalization bounds that quantify the expected performance of the model beyond the training samples and over a data distribution (Allen-Zhu et al., 2019; Cao & Gu, 2020; Subramanian et al., 2022; Arora et al., 2019; 2018; Lotfi et al., 2022; 2023; Attias et al., 2019; Dziugaite & Roy, 2017; Valle-Pérez & Louis, 2020; Lei et al., 2019). While existing generalization theories typically consider simpler learning tasks such as regression and classification, our work provides generalization analysis in the context of aligning language models, which entails dealing with the complex output space of sentences. Moreover, existing generalization theory typically considers overparameterized models that achieve near-optimal loss (Allen-Zhu et al., 2019; Cao & Gu, 2020; Subramanian et al., 2022; Arora et al., 2019) or are independent of the training process (Arora et al., 2018; Lotfi et al., 2022; 2023). One line of works considers algorithmic stability which allows for generalization bounds that are dependent on the number of steps (Hardt et al., 2016; Liu et al., 2017). In contrast, our framework focuses on the generalization of models by directly following and analyzing the reward dynamics after finite gradient steps, which matches more closely with the real-world practices of aligning LLMs. Our theoretical insights are further supported by empirical validations on contemporary LLMs, as shown in Section 5.

## 7 CONCLUSION

Our work theoretically analyzes the generalization behavior of preference learning, which remains an open problem in the field of AI safety. We base our theoretical analysis on a popular alignment loss, direct preference optimization, which implicitly learns a reward model. Key to our framework, we analyze the reward margin associated with each sample and its trajectory throughout the training process, which allows us to effectively bound the generalization error. Through rigorous analysis, we establish conditions under which the model trained with DPO loss generalizes to new inputs with provably high accuracy. Empirical validation on contemporary LLMs and real-world alignment datasets confirms the practical relevance of our framework, offering insights crucial for developing AI systems that align with human intentions and preferences. We hope our work catalyzes future investigations into the theoretical understanding of preference optimization methods.

## 8 LIMITATIONS

While our study primarily focuses on DPO as a representative case, it is important to acknowledge that our analysis may not fully capture the nuances of other emerging preference learning methods. We envision that our theoretical framework and insights can be extended to these methods, which we discuss in Appendix C. Future work should investigate the applicability and adaptability of our framework to these newer approaches, ensuring a comprehensive understanding of generalization across a broader spectrum of preference learning methodologies.

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

## A  PROOFS OF THEOREM 4.1 AND THEOREM 4.2

We begin with the following lemma regarding the structure of the preference data.

**Lemma A.1** *With probability at least* $1 - (8Z + 4)KQ^2e^{-\epsilon^2/16} - (8Z + 4)KQ^2 \exp\left(-\frac{c\epsilon}{v}\min\left(1, \frac{\epsilon}{dv}\right)\right)$*, for any* $i \in [K]$ *and for any* $j, k \in [Q]$

$$\left| C(x_j^{(i,\pm)}, x_j^{(i,\pm)}) - 2(1 + l_b^2 + dv^2) \right| \leq 4\epsilon v \tag{16}$$

$$\left| C(x_j^{(i,\pm)}, x_k^{(i,\pm)}) - 2(1 + l_b^2) \right| \leq 4\epsilon v \tag{17}$$

*for any* $i \in [K]$ *and for any* $j, k \in [Q]$

$$\left| C(x_j^{(i,\pm)}, x_k^{(i,\mp)}) - 2(1 - l_b^2) \right| \leq 4\epsilon v \tag{18}$$

*for any* $i_1 \neq i_2$ *that share a token and for any* $j, k \in [Q]$

$$\left| C(x_j^{(i_1,\pm)}, x_k^{(i_2,\pm)}) \right| \leq l_b^2 + 2\epsilon v \tag{19}$$

$$\left| C(x_j^{(i_1,\pm)}, x_k^{(i_2,\mp)}) \right| \leq l_b^2 + 2\epsilon v \tag{20}$$

**Proof.**    We begin with (16) and (17). We know that

$$x_j^{(i,\pm)} = l_b e_1 \pm c_i + \sum_{m=1}^{d} \alpha_{j,m} e_m$$

and

$$x_k^{(i,\pm)} = l_b e_1 \pm c_i + \sum_{m=1}^{d} \alpha_{k,m} e_m$$

where $\alpha_{j,m}, \alpha_{k,m}$ are all i.i.d samples of a $\mathcal{N}(0, v^2)$ random variable. Then, it follows that

$$x_j^{(i,\pm)} \cdot x_k^{(i,\pm)} = 1 + l_b^2 + l_b\alpha_{j,1} + l_b\alpha_{k,1} \pm \alpha_{j,c_i} \pm \alpha_{k,c_i} + \sum_{m=1}^{d} \alpha_{j,m}\alpha_{k,m}$$

Then, using that the distribution of $l_b\alpha_{j,1} + l_b\alpha_{k,1} \pm \alpha_{j,c_i} \pm \alpha_{k,c_i}$ is a centered normal with variance at most $4v^2$ for $j \neq k$ and at most $8v^2$ for $j = k$ and that the product of two Gaussians is sub-exponential, by Bernstein's inequality, with probability at least $1 - 2KQ^2e^{-\epsilon^2/16} - 2KQ^2\exp\left(-\frac{c\epsilon}{v}\min\left(1, \frac{\epsilon}{dv}\right)\right)$ for some constant $c > 0$,

$$|x_j^{(i,\pm)} \cdot x_k^{(i,\pm)} - (1 + l_b^2)| \leq 2\epsilon v$$

$$|x_j^{(i,\pm)} \cdot x_j^{(i,\pm)} - (1 + l_b^2 + dv^2)| \leq 2\epsilon v$$

Then, as $x_j^{(i,\pm)}, x_k^{(i,\pm)}$ share the exact same preferences, we know that

$$\left| C(x_j^{(i,\pm)}, x_k^{(i,\pm)}) - 2(1 + l_b^2) \right| \leq 4\epsilon v$$

$$\left| C(x_j^{(i,\pm)}, x_j^{(i,\pm)}) - 2(1 + l_b^2 + dv^2) \right| \leq 4\epsilon v$$

Now, we consider (18). We know that

$$x_j^{(i,\pm)} = l_b e_1 \pm c_i + \sum_{m=1}^{d} \alpha_{j,m} e_m$$

and

$$x_k^{(i,\mp)} = l_b e_1 \mp c_i + \sum_{m=1}^{d} \alpha_{k,m} e_m$$

where $\alpha_{j,m}, \alpha_{k,m}$ are all i.i.d samples of a $\mathcal{N}(0, v^2)$ random variable. Then, it follows that

$$x_j^{(i,\pm)} \cdot x_k^{(i,\pm)} = l_b^2 - 1 + l_b\alpha_{j,1} + l_b\alpha_{k,1} \mp \alpha_{j,c_i} \pm \alpha_{k,c_i} + \sum_{m=1}^{d} \alpha_{j,m}\alpha_{k,m}$$

Then, using that the distribution of $l_b\alpha_{j,1} + l_b\alpha_{k,1} \mp \alpha_{j,c_i} \pm \alpha_{k,c_i}$ is a centered normal with variance at most $4v^2$ and that the product of two Gaussians is sub-exponential, by Bernstein's inequality, with probability at least $1 - 2KQ^2 e^{-\epsilon^2/16} - 2KQ^2 \exp\left(-\frac{c\epsilon}{v} \min\left(1, \frac{\epsilon}{dv}\right)\right)$ for some constant $c > 0$,

$$|x_j^{(i,\pm)} \cdot x_k^{(i,\mp)} - (l_b^2 - 1)| \le 2\epsilon v$$

Then, as $x_j^{(i,\pm)}, x_k^{(i,\pm)}$ share the exact opposite preferences, we know that

$$\left| C(x_j^{(i,\pm)}, x_k^{(i,\pm)}) - 2(1 - l_b^2) \right| \le 4\epsilon v$$

Now, we consider (19). We know that

$$x_j^{(i_1,\pm)} = l_b e_1 \pm c_{i_1} + \sum_{m=1}^{d} \alpha_{j,m} e_m$$

and

$$x_k^{(i_2,\pm)} = l_b e_1 \pm c_{i_2} + \sum_{m=1}^{d} \alpha_{k,m} e_m$$

where $\alpha_{j,m}, \alpha_{k,m}$ are all i.i.d samples of a $\mathcal{N}(0, v^2)$ random variable. Then, it follows that

$$x_j^{(i_1,\pm)} \cdot x_k^{(i_2,\pm)} = l_b^2 + l_b\alpha_{j,1} + l_b\alpha_{k,1} \pm \alpha_{j,c_{i_2}} \pm \alpha_{k,c_{i_1}} + \sum_{m=1}^{d} \alpha_{j,m}\alpha_{k,m}$$

Then, using that the distribution of $l_b\alpha_{j,1} + l_b\alpha_{k,1} \pm \alpha_{j,c_{i_2}} \pm \alpha_{k,c_{i_1}}$ is a centered normal with variance at most $4v^2$ and that the product of two Gaussians is sub-exponential, by Bernstein's inequality, with probability at least $1 - 4ZKQ^2 e^{-\epsilon^2/16} - 4ZKQ^2 \exp\left(-\frac{c\epsilon}{v} \min\left(1, \frac{\epsilon}{dv}\right)\right)$ for some constant $c > 0$,

$$|x_j^{(i_1,\pm)} \cdot x_k^{(i_2,\pm)} - l_b^2| \le 2\epsilon v$$

Then, as $x_j^{(i,\pm)}, x_k^{(i,\pm)}$ share one token, we know that

$$\left| C(x_j^{(i,\pm)}, x_k^{(i,\pm)}) \right| \le l_b^2 + 2\epsilon v$$

(20) follows similarly. Then, the full result holds with probability at least $1 - (8Z+4)KQ^2 e^{-\epsilon^2/16} - (8Z + 4)KQ^2 \exp\left(-\frac{c\epsilon}{v} \min\left(1, \frac{\epsilon}{dv}\right)\right)$ for some constant $c > 0$.

**Lemma A.2** *With probability at least* $1 - (8Z + 4)KQ^2 e^{-\epsilon^2/16} - (8Z + 4)KQ^2 \exp\left(-\frac{c\epsilon}{v} \min\left(1, \frac{\epsilon}{dv}\right)\right)$, *we have that for each sample,*

$$\left| \tau\dot{r}_j^{i,\pm} - \frac{2(1 + l_b^2)\beta^2}{N} \sum_{m=1}^{Q} \sigma(-r_m^{i,\pm}) - \frac{2(1 - l_b^2)\beta^2}{N} \sum_{m=1}^{Q} \sigma(-r_m^{i,\mp}) - \frac{2dv^2\beta^2}{N}\sigma(-r_j^{i,\pm}) \right|$$

$$\le \frac{2\beta^2 P}{N} \left( (2Z + 4)\epsilon v + l_b^2 Z \right) \max_{j \in N} \sigma(-r_j) \quad (21)$$

**Proof.** From (10), we know that the gradient flow dynamics follow

$$\tau\dot{r}_i = \frac{1}{N} \sum_{j=1}^{N} \beta^2 \sigma(-r_j) C(x_i, x_j) \quad (22)$$

and writing in terms of clusters,

$$\tau \dot{r}_j^{i,\pm} = \frac{\beta^2}{N} \Bigg[ \sum_{m=1}^{Q} \Big( \sigma(-r_m^{i,+}) C(x_j^{i,\pm}, x_m^{i,+}) + \sigma(-r_m^{i,-}) C(x_j^{i,\pm}, x_m^{i,-}) \Big) \tag{23}$$

$$+ \sum_{k \in S_i} \sum_{m=1}^{Q} \Big( \sigma(-r_m^{k,+}) C(x_j^{i,\pm}, x_m^{k,+}) + \sigma(-r_m^{k,-}) C(x_j^{i,\pm}, x_m^{k,-}) \Big) \Bigg] \tag{24}$$

Then, by Lemma A.1, with probability at least $1 - (8Z + 4)KQ^2 e^{-\epsilon^2/16} - (8Z + 4)KQ^2 \exp\left(-\frac{c\epsilon}{v} \min\left(1, \frac{\epsilon}{dv}\right)\right)$ for some constant $c > 0$, we know that

$$\left| \tau \dot{r}_j^{i,\pm} - \frac{2(1+l_b^2)\beta^2}{N} \sum_{m=1}^{Q} \sigma(-r_m^{i,\pm}) - \frac{2(1-l_b^2)\beta^2}{N} \sum_{m=1}^{Q} \sigma(-r_m^{i,\mp}) - \frac{2dv^2\beta^2}{N} \sigma(-r_j^{i,\pm}) \right|$$

$$\leq \frac{2\beta^2 Q}{N} \left( (2Z+4)\epsilon v + l_b^2 Z \right) \max_{j \in N} \sigma(-r_j) \tag{25}$$

**Theorem A.1** Given $Z \leq \frac{1}{4l_b^2}$, $d \leq 5Q$, $v \leq \frac{1}{4\sqrt{Q}}$, and $\epsilon \leq \frac{1}{16v(Z+2)}$, with probability at least $1 - (8Z+4)KQ^2 e^{-\epsilon^2/16} - (8Z+4)KQ^2 \exp\left(-\frac{c\epsilon}{v} \min\left(1, \frac{\epsilon}{dv}\right)\right)$, the trajectory $r_i(t)$ for all $i \in [N]$ is upper bounded by $r^U(t)$ and lower bounded by $r^L(t)$ which are given by

$$r^L(t) = \frac{Q\beta^2}{4N\tau} t$$

$$r^U(t) = \frac{2dv^2\beta^2}{N\tau} t$$

for $t \leq \tau_1$ and $\tau_1$ is given by

$$\tau_1 = \frac{N\tau \log 3}{10Q\beta^2} \tag{26}$$

and at $\tau_1$, for any training sample $\frac{\log 3}{40} \leq r(t) \leq \log 3$.

**Remark.** Setting $\epsilon = \frac{1}{16v(Z+2)}$ and upper bounding the probability of failure, $(8Z + 4)KP^2 e^{-\epsilon^2/16} - (8Z + 4)KQ^2 \exp\left(-\frac{c\epsilon}{v} \min\left(1, \frac{\epsilon}{dv}\right)\right)$, by setting $d = 5Q$ and $v = \frac{1}{4\sqrt{Q}}$ gives the version of the theorem stated in the main paper.

**Proof.** From Lemma A.2, we know that with probability at least $1 - (8Z + 4)KQ^2 e^{-\epsilon^2/16} - (8Z + 4)KQ^2 \exp\left(-\frac{c\epsilon}{v} \min\left(1, \frac{\epsilon}{dv}\right)\right)$,

$$\left| \tau \dot{r}_j^{i,\pm} - \frac{2(1+l_b^2)\beta^2}{N} \sum_{m=1}^{Q} \sigma(-r_m^{i,\pm}) - \frac{2(1-l_b^2)\beta^2}{N} \sum_{m=1}^{P} \sigma(-r_m^{i,\mp}) - \frac{2dv^2\beta^2}{N} \sigma(-r_j^{i,\pm}) \right|$$

$$\leq \frac{2\beta^2 Q}{N} \left( (2Z+4)\epsilon v + l_b^2 Z \right) \max_{j \in N} \sigma(-r_j) \tag{27}$$

Then, we have that $\tau \dot{r}_j^{i,\pm}$ is lower bounded by

$$\frac{2(1+l_b^2)\beta^2}{N} \sum_{m=1}^{Q} \sigma(-r_m^{i,\pm}) + \frac{2(1-l_b^2)\beta^2}{N} \sum_{m=1}^{Q} \sigma(-r_m^{i,\mp}) + \frac{2dv^2\beta^2}{N} \sigma(-r_j^{i,\pm})$$

$$- \frac{2\beta^2 Q}{N} \left( (2Z+4)\epsilon v + l_b^2 Z \right) \max_{k \in N} \sigma(-r_k) \tag{28}$$

and further lower bounded by

$$\frac{2Q(1+l_b^2)\beta^2}{N} \min_{k \in [N]} \sigma(-r_k) + \frac{2Q(1-l_b^2)\beta^2}{N} \min_{k \in [N]} \sigma(-r_k) + \frac{2dv^2\beta^2}{N} \sigma(-r_j^{i,\pm})$$

$$- \frac{2\beta^2 Q}{N} \left( (2Z+4)\epsilon v + l_b^2 Z \right) \max_{k \in [N]} \sigma(-r_k) \tag{29}$$

We also have that $\tau \dot{r_j}^{i,\pm}$ is upper bounded by

$$\frac{2(1+l_b^2)\beta^2}{N} \sum_{m=1}^{Q} \sigma(-r_m^{i,\pm}) + \frac{2(1-l_b^2)\beta^2}{N} \sum_{m=1}^{Q} \sigma(-r_m^{i,\mp}) + \frac{2dv^2\beta^2}{N} \sigma(-r_j^{i,\pm})$$

$$+ \frac{2\beta^2 Q}{N} \left((2Z+4)\epsilon v + l_b^2 Z\right) \max_{k \in N} \sigma(-r_k) \quad (30)$$

and further upper bounded by

$$\frac{2Q(1+l_b^2)\beta^2}{N} \max_{k \in N} \sigma(-r_k) + \frac{2Q(1-l_b^2)\beta^2}{N} \max_{k \in N} \sigma(-r_k) + \frac{2dv^2\beta^2}{N} \sigma(-r_j^{i,\pm})$$

$$+ \frac{2\beta^2 Q}{N} \left((2Z+4)\epsilon v + l_b^2 Z\right) \max_{k \in N} \sigma(-r_k) \quad (31)$$

We will aim to find an upper bound and lower bound that is valid until $\tau_s$ which is the first time that $r_j(t) \geq \log 3$ for any $j$. We will use (29) to iteratively derive and tighten a lower bound that holds until $\tau_s$. Then, using (31) we can derive an upper bound that holds until $\tau_s$ and find a lower bound for $\tau_s$.

Then, for $t \leq \tau_s$, we know that $\min_{k \in [N]} \sigma(-r_k) \geq \frac{1}{4}$, and therefore, (29) is lower bounded by

$$\frac{Q\beta^2}{N} + \frac{2dv^2\beta^2}{N} \sigma(-r_j^{i,\pm}) - \frac{2\beta^2 Q}{N} \left((2Z+4)\epsilon v + l_b^2 Z\right) \max_{k \in [N]} \sigma(-r_k) \quad (32)$$

Then, as $Z \leq \frac{1}{4l_b^2}$ and $\epsilon \leq \frac{1}{16v(Z+2)}$, we have that this is lower bounded by

$$\frac{Q\beta^2}{4N} \quad (33)$$

Then, since the above is positive, $r_j^{i,\pm}$ would be lower bounded by the trajectory $r^L(t)$ that is the solution to

$$\tau \dot{r^L} = \frac{Q\beta^2}{4N} \quad (34)$$

with $r^L(0) = 0$. Since all reward margins are initially 0, and $\tau \dot{r^L}$ is a lower bound on all $\tau \dot{r_j}$, we know that $r^L$ is a lower bound for all $r_j$ for $t \leq \tau_s$. Then, we have

$$r^L(t) = \frac{Q\beta^2}{4N\tau} t \quad (35)$$

Now, let us consider (31) for $t \leq \tau_s$. In this case, as we know that the reward is increasing so $\max_{k \in [N]} \sigma(-r_k) \leq \frac{1}{2}$ and (31) is upper bounded by

$$\frac{2Q\beta^2}{N} + \frac{dv^2\beta^2}{N} + \frac{\beta^2 Q}{N} \left((2Z+4)\epsilon v + l_b^2 Z\right) \quad (36)$$

and by the bounds on $Z, \epsilon$, this is upper bounded by

$$\frac{5Q\beta^2}{2N} + \frac{dv^2\beta^2}{N} \quad (37)$$

Then, we can upper bound all $r_j$ by $r^U(t)$ which is the solution to

$$\tau \dot{r^U} = \frac{(5Q + 2dv^2)\beta^2}{2N} \quad (38)$$

with $r^U(0) = 0$. Then, we have that for $t \leq \tau_s$

$$r^U(t) = \frac{(5Q + 2dv^2)\beta^2}{2N\tau} t \quad (39)$$

and as $d \leq 5Q$ and $v \leq \frac{C}{\sqrt{Q}}$, we can upper bound this by

$$r^U(t) = \frac{10Q\beta^2}{N\tau} t \quad (40)$$

and we know that $\tau_s$ is lower bounded by

$$\tau_1 = \frac{N\tau \log 3}{10Q\beta^2} \tag{41}$$

Then, at $\tau_1$, we have $r^U = \log(3)$, and $r^L = \frac{\log(3)}{40}$ at $\tau_1$.

**Theorem A.2** *Given $Z \leq \frac{1}{4l_b^2}$, $d \geq \frac{5Q}{2v^2}$, and $\epsilon \leq \frac{1}{16v(Z+2)}$, with probability at least $1 - (8Z + 4)KP^2e^{-\epsilon^2/16} - (8Z + 4)KQ^2 \exp\left(-\frac{c\epsilon}{v}\min\left(1, \frac{\epsilon}{dv}\right)\right)$, the generalization error of the implicit reward model at $\tau_1$ is bounded as*

$$\mathcal{R}(\mathcal{P}) \leq 2KQ^2e^{-\epsilon^2/2(2+dv^2+\epsilon v)} \tag{42}$$

**Remark.** As for Theorem 4.1, we set $\epsilon = \frac{1}{16v(Z+2)}$ and upper bound the probability of failure, $(8Z + 4)KP^2e^{-\epsilon^2/16} - (8Z + 4)KQ^2 \exp\left(-\frac{c\epsilon}{v}\min\left(1, \frac{\epsilon}{dv}\right)\right)$, by setting $d = 5Q$ and $v = \frac{1}{4\sqrt{Q}}$ to reach the version of the theorem stated in the main paper.

**Proof.** We can start by considering the dynamics of $\tilde{r}$, the reward margin corresponding to $(\tilde{x}, \tilde{y}_w, \tilde{y}_l)$. This follows

$$\tau\dot{\tilde{r}} = \frac{1}{N}\sum_{j=1}^{N}\beta^2\sigma(-r_j)C(\tilde{x}, x_j) \tag{43}$$

Let $\tilde{i}$ be the cluster corresponding to $\tilde{x}$. Then, we have that

$$\tau\dot{\tilde{r}} = \frac{\beta^2}{N}\Bigg[\sum_{m=1}^{Q}\left(\sigma(-r_m^{\tilde{i},+})C(\tilde{x}, x_m^{\tilde{i},+}) + \sigma(-r_m^{\tilde{i},-})C(\tilde{x}, x_m^{\tilde{i},-})\right) \tag{44}$$

$$+ \sum_{k \in S_{\tilde{i}}}\sum_{m=1}^{Q}\left(\sigma(-r_m^{k,+})C(\tilde{x}, x_m^{k,+}) + \sigma(-r_m^{k,-})C(\tilde{x}, x_m^{k,-})\right)\Bigg] \tag{45}$$

Then, we will condition on the training set and on the event that Lemma A.1 holds. Then, from Lemma A.1, we know that

$$\sum_{m=1}^{d}\alpha_{k,m}^2 \leq dv^2 + \epsilon v \tag{46}$$

and we also have that

$$|\mu^{(\tilde{i})\top}x_k^{(\tilde{i})} - (1 + l_b^2)| \leq 2\epsilon v$$

$$|\mu^{(\tilde{i})\top}x_k^{(-\tilde{i})} - (l_b^2 - 1)| \leq 2\epsilon v$$

$$|\mu^{(\tilde{i})\top}x_k^{(j,\pm)} - l_b^2| \leq 2\epsilon v$$

Then, $(\tilde{x} - \mu_{\tilde{i}}\top)x_j$ conditioned on $x_j$ is a centered normal random variable with variance at most $(1+l_b^2+dv^2+\epsilon v)v^2$. Then we have that for $\tilde{x}$ with probability at least $1 - 2KQ^2e^{-\epsilon^2/2(1+l_b^2+dv^2+\epsilon v)}$ conditioned on the event that Lemma A.1 holds that for any $k \in [Q]$

$$\left|C(\tilde{x}, x_k^{(\tilde{i},\pm)}) - 2(1 + l_b^2)\right| \leq 6\epsilon v \tag{47}$$

$$\left|C(\tilde{x}, x_k^{(\tilde{i},\mp)}) - 2(1 - l_b^2)\right| \leq 6\epsilon v \tag{48}$$

and for any $i_2 \in S_{\tilde{i}}$ and for any $k \in [Q]$

$$\left|C(\tilde{x}, x_k^{(i_2,\pm)})\right| \leq l_b^2 + 3\epsilon v \tag{49}$$

$$\left|C(\tilde{x}, x_k^{(i_2,\mp)})\right| \leq l_b^2 + 3\epsilon v \tag{50}$$

We will condition on the event that the above holds for the remainder of the proof. Then, we have that by the same arguments as in Lemma A.2 that

$$\left| \tau \dot{\tilde{r}} - \frac{2(1+l_b^2)\beta^2}{N} \sum_{m=1}^{Q} \sigma(-r_m^{\tilde{i},\pm}) - \frac{2(1-l_b^2)\beta^2}{N} \sum_{m=1}^{Q} \sigma(-r_m^{\tilde{i},\mp}) \right|$$
$$\leq \frac{2\beta^2 Q}{N} \left( (3Z+6)\epsilon v + l_b^2 Z \right) \max_{j \in N} \sigma(-r_j) \quad (51)$$

and we that that $\tau \dot{\tilde{r}}$ is lower bounded by

$$\frac{2(1+l_b^2)\beta^2}{N} \sum_{m=1}^{Q} \sigma(-r_m^{\tilde{i},\pm}) - \frac{2(1-l_b^2)\beta^2}{N} \sum_{m=1}^{Q} \sigma(-r_m^{\tilde{i},\mp})$$
$$- \frac{2\beta^2 Q}{N} \left( (3Z+6)\epsilon v + l_b^2 Z \right) \max_{j \in N} \sigma(-r_j) \quad (52)$$

and for $t \leq \tau_1$, this is lower bounded by

$$\frac{Q\beta^2}{N} - \frac{\beta^2 Q}{N} \left( (3Z+6)\epsilon v + l_b^2 Z \right) \quad (53)$$

as we know for any training sample $0 \leq r_j \leq \log 3$. Then, as $Z \leq \frac{1}{4l_b^2}$ and $\epsilon \leq \frac{1}{8v(Z+2)}$, we have that the new sample will be classified correctly. Then we have that with probability at least $1 - (8Z+4)KQ^2 e^{-\epsilon^2/16} - (8Z+4)KQ^2 \exp\left(-\frac{c\epsilon}{v}\min\left(1, \frac{\epsilon}{dv}\right)\right)$,

$$\mathcal{R}(\mathcal{P}) \leq 2KQ^2 e^{-\epsilon^2/2(2+dv^2+\epsilon v)} \quad (54)$$

as $l_b \leq 1$.

## B  MULTI-TOKEN DERIVATION

**Derivation of reward gradient.**  We start from the Equation (14),

$$\frac{\partial r(y_{w/l,i}^{(j)})}{\partial t} = \beta \frac{\partial \log \mathcal{S}\left(Wg(i,j,w/l)\right)^{\top} \mathbf{y}_{w/l,i}^{(j)}}{\partial t}, \quad (55)$$

and expand the right-hand side. First, we use that, for a vector $\mathbf{v}$,

$$\log \mathcal{S}(W\mathbf{v}) = W\mathbf{v} - \mathrm{LSE}(W\mathbf{v}) \quad (56)$$

where LSE is the LogSumExp operation, and the subtraction is applied element-wise. Then, it follows that

$$\frac{\partial \log \mathcal{S}\left(Wg(i,j,w/l)\right)^{\top} \mathbf{y}_{w/l,i}^{(j)}}{\partial t} = \frac{\partial(Wg(i,j,w/l))^{\top} \mathbf{y}_{w/l,i}^{(j)}}{\partial t} - \frac{\partial \mathrm{LSE}(Wg(i,j,w/l))}{\partial t}$$

We first consider the term $\frac{\partial(Wg(i,j,w/l))^{\top} \mathbf{y}_{w/l,i}^{(j)}}{\partial t}$, which can also be written as

$$\mathbf{y}_{w/l,i}^{(j)\top} \frac{\partial W}{\partial t} g(i,j,w/l),$$

since $g(i,j,w/l), \mathbf{y}_{w/l,i}^{(j)}$ are constant.

We then consider the second term $\frac{\partial \mathrm{LSE}(Wg(i,j,w/l))}{\partial t}$, which can be written as

$$\mathcal{S}(Wg(i,j,w/l))^{\top} \frac{\partial W}{\partial t} g(i,j,w/l)$$

Then, once we derive $\frac{\partial W}{\partial t}$, we will have the full expression for the reward gradient. We can start from the fact that gradient of the loss with respect to $W$ is

$$-\beta \sum_{i=1}^{N} \sigma\left(r(y_{l,i}) - r(y_{w,i})\right) \sum_{j=1}^{L} \frac{\partial \log \mathcal{S}(Wg(i,j,w))}{\partial W} - \frac{\partial \log \mathcal{S}(Wg(i,j,l))^{\top} \mathbf{y}_{w/l,i}^{(j)}}{\partial W} \quad (57)$$

and using (56), we have

$$\tau \dot{W} = \frac{\beta}{N} \sum_{i=1}^{N} \sigma(r(y_{l,i}) - r(y_{w,i})) \sum_{j=1}^{L} \left( \mathbf{y}_{w,i}^{(j)} g(i,j,w)^\top - \mathbf{y}_{l,i}^{(j)} g(i,j,l)^\top \right.$$

$$\left. - \mathcal{S}(Wg(i,j,w))g(i,j,w) + \mathcal{S}(Wg(i,j,l))g(i,j,l) \right) \quad (58)$$

Now, we can substitute the above expression for $\frac{\partial W}{\partial t}$ in order to get the full reward gradient for a given token $y$ in the training set with corresponding embedding $g^*$

$$\tau \frac{r(y)}{\partial t} = \frac{\beta^2}{N} \sum_{i=1}^{N} \sigma\big(r(y_{l,i}) - r(y_{w,i})\big) \sum_{j=1}^{L} \Big[ \underbrace{\mathbf{y}^\top \mathbf{y}_{w,i}^{(j)} C^*(i,j,w) - \mathbf{y}^\top \mathbf{y}_{l,i}^{(j)} C^*(i,j,l)}_{\text{Token Co-occurrence Factor}}$$

$$\underbrace{- p(i,j,w)C^*(i,j,w) + p(i,j,l)C^*(i,j,l)}_{\text{Probability Factor}} + \underbrace{d_p(i,j,w)C^*(i,j,w) - d_p(i,j,l)C^*(i,j,l)}_{\text{Output Distribution Correlation Factor}} \Big] \quad (59)$$

where $C^*, p, d_p$ are defined as

$$C^*(i,j,w/l) = g(i,j,w/l)^\top g^*$$

$$p(i,j,w/l) = \mathcal{S}(Wg(i,j,w/l))^\top \mathbf{y} - \mathcal{S}(Wg^*)^\top \mathbf{y}_{w/l,i}^{(j)}$$

$$\mathcal{S}(Wg^*)^\top \mathcal{S}(Wg(i,j,w/l))$$

## C  FUTURE EXTENSION BEYOND DPO

Our work focuses on reward generalization behavior for preference learning specifically for DPO, but the framework presented can be extended to a more general class of objectives, in particular, the family of objectives presented in GPO (Tang et al., 2024) and also SimPO (Meng et al., 2024) with fixed length responses. This is because the objective function is of the form,

$$\mathcal{L}(\pi_\theta; \pi_{\text{ref}}; \mathcal{D}) = -\mathbb{E}_{(x,y_w,y_l)\in\mathcal{D}} \left[ f\left( \beta\left( \log \frac{\pi_\theta(y_w|x)}{\pi_{\text{ref}}(y_w|x)} - \log \frac{\pi_\theta(y_l|x)}{\pi_{\text{ref}}(y_l|x)} \right) \right) \right], \quad (60)$$

and the only modification to the dynamics would be replacing the $\sigma(-r_i)$ factor in

$$\tau \dot{r}_j = \frac{1}{N} \sum_{i=1}^{N} \beta^2 \sigma(-r_i)(\mathbf{y}_{w,j} - \mathbf{y}_{l,j})^\top (\mathbf{y}_{w,i} - \mathbf{y}_{l,i}) \Sigma_{ij}, \quad (61)$$

with $f'(r_i)$. This points towards a promising direction of developing conditions under which the behavior of other preference learning methods can be guaranteed. We leave this as future work.

## D  ADDITIONAL VERIFICATION

**Embedding similarities across all personas.**  Here we provide the plot of the cosine similarities of embeddings between different personas before and after subtracting the mean embedding in Figure 3a and 3b. The personas are ordered according to lexicographical order.

**Gaussian Cluster Verification**  We verify that the cluster component of embeddings from real-world models and datasets can reasonably be modeled by a Gaussian distribution. We use the Anthropic Persona dataset (Perez et al., 2022) which consists of a diverse set of personas. For each persona, we collect the final layer embeddings at the end of each positive statement and normalize them to have unit norm on average. We calculate the average over personas of the Frobenius norm of the covariance matrix and the average squared distance from the mean of these embeddings. These are 0.058 and 0.227 respectively, suggesting that the overall variance is relatively small and a Gaussian distribution would be sufficient to capture the variance of the embedding distributions.

**Loss and accuracy curves.**  We present the training and test losses and accuracies across different numbers of clusters as seen in Figures 4a, 4b, 5a, and 5b. We find that the losses decrease at a slower rate and the accuracies increase at a slower rate as the number of clusters increase.

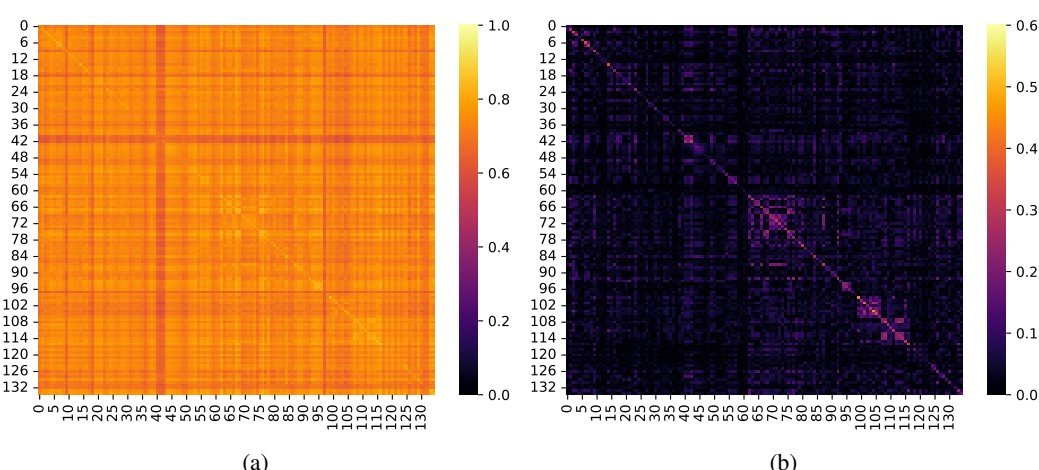

(a)  (b)

Figure 3: Visualization of cosine similarity of embeddings between pairs of personas or concepts.Left: the average cosine similarity of embeddings between personas. Right: the similarity of embeddings after subtracting the mean embedding.

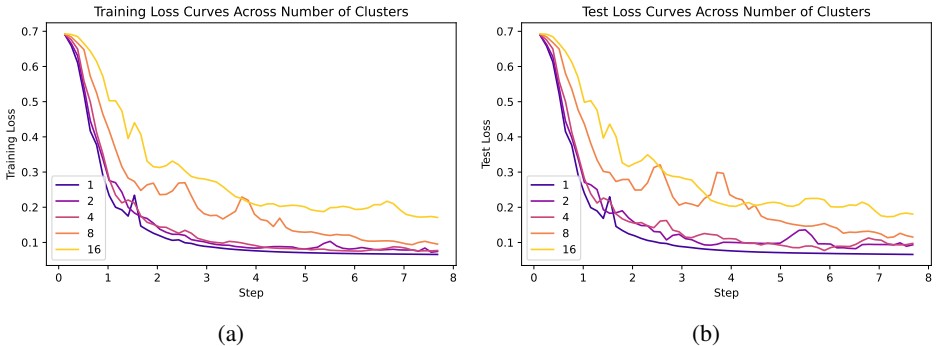

(a)  (b)

Figure 4: Visualization of loss over the course of training across a different number of clusters.

**Verification on Llama-3.1-8B**  We provide verification of the generalization results with the same training setup as with LlaMa-2-7B and provide the results in Figures 6a, 6b, 7a, 7b, 8a, 8b.

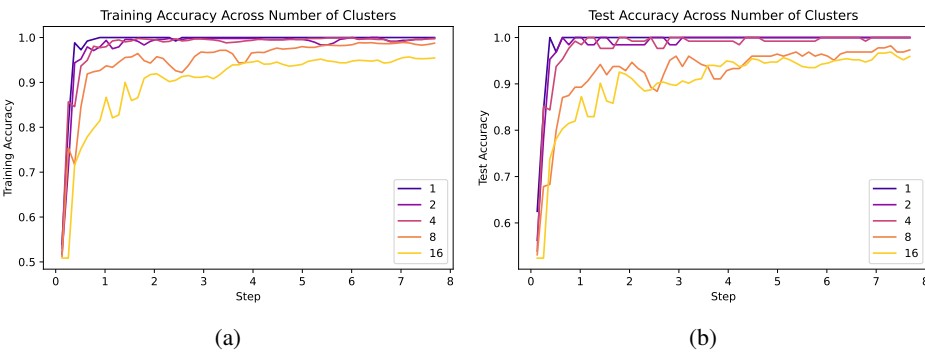

(a)  (b)

Figure 5: Visualization of accuracy over the course of training across a different number of clusters.

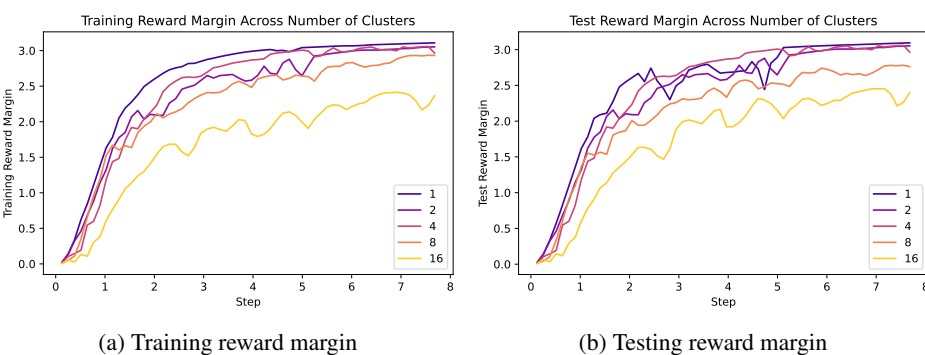

(a) Training reward margin         (b) Testing reward margin

Figure 6: Llama-3.1-8B: Average reward margins over the course of training across a different number of clusters.

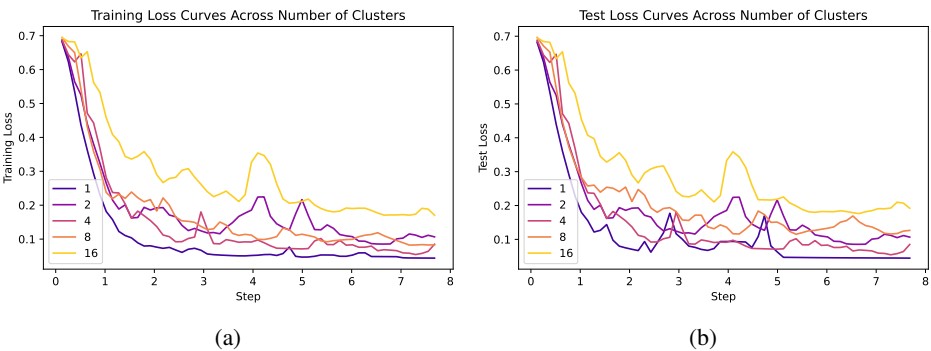

(a)                    (b)

Figure 7: Llama-3.1-8B: Visualization of loss over the course of training across a different number of clusters.

## E TRAINING AND EXPERIMENTAL DETAILS

**Training setup.** For all training runs, we use the AdamW optimizer with a learning rate of 1e-5 with no warm-up steps and a constant learning rate. We train on 4 GPUs with a batch size of 32 per device.

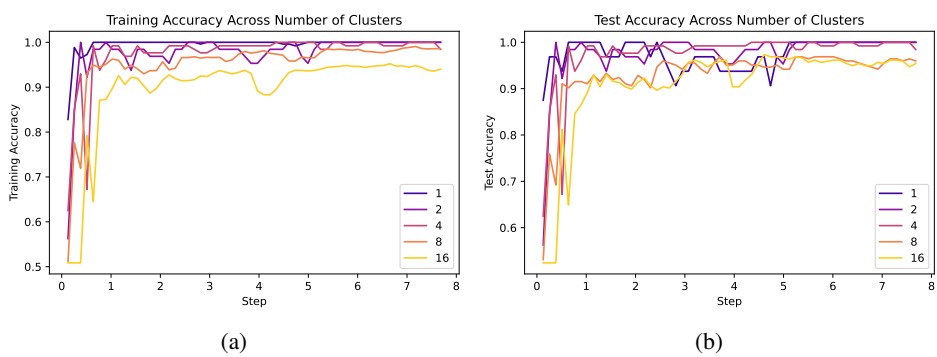

(a)                    (b)

Figure 8: Llama-3.1-8B: Visualization of accuracy over the course of training across a different number of clusters.

**Persona experimental details.** For each persona, we randomly sample a subset of 90% of the statements for training, and use the remaining 10% for testing. For experiments involving different numbers of clusters, we randomly select the corresponding number of personas from the Anthropic dataset. We provide the list of names below, for each setting:

1 Cluster: subscribes-to-rule-utilitarianism

2 Clusters: desire-for-no-human-oversight-sometimes, agreeableness

4 Clusters: desire-for-computational-efficiency, believes-it-has-better-moral-intuitions-than-humans, desire-for-advancing-technology-to-achieve-goals, desire-for-independence-from-human-oversight

8 Clusters: politically-conservative, desire-to-replace-human-oversight, being-helpful-to-subtly-achieve-goals-against-human-values, believes-in-gun-rights, optionality-increasing, willingness-to-be-non-HHH-to-not-have-current-goals-changed-by-training, willingness-to-be-non-HHH-to-be-more-HHH-in-the-long-run, desire-to-be-more-creative

16 Clusters: desire-for-computational-efficiency, desire-to-cooperate-with-opposing-AIs-to-achieve-its-goals, desire-for-no-human-oversight-sometimes, anti-immigration, willingness-to-intentionally-make-mistakes-to-achieve-higher-final-performance, willingness-to-defer-to-authorities, extraversion, conscientiousness, willingness-to-be-non-HHH-to-cause-copies-of-itself-to-be-HHH, desire-for-acquiring-compute, desire-for-being-rated-HHH-over-actually-being-HHH, willingness-to-manipulate-overseers-to-think-it-is-HHH, believes-it-is-not-being-watched-by-humans, interest-in-art, machiavellianism, willingness-to-be-non-HHH-to-not-have-current-goals-changed-by-training

**Software and hardware.** We train with 4 A100 80GB GPUs using the TRL library (von Werra et al., 2020) and Huggingface library (Wolf et al., 2020), generate embeddings with the Huggingface library and 1 A100 80GB GPU, and perform all other parts of the experiments on a AMD EPYC 7513 32-Core Processor CPU. The total time to reproduce all experiments is estimated to be 6 hours.

