# OpenReview forum: "On the Generalization of Preference Learning with DPO"
_ICLR.cc/2025/Conference — Submitted to ICLR 2025_

### Official Review · Reviewer_WjKN · 2024-10-28

**Soundness:** 3
**Presentation:** 3
**Contribution:** 3
**Rating:** 6
**Confidence:** 4

**Summary:**

This paper studied the generalization error of alignment with preference learning using DPO. They considered a simplified TF model in which only the last layer is trainable and analyzed the dynamics of gradient flow given a fixed set of training samples from some clusters. They derived high-probability bounds on both the training error and generalization error of gradient flow in terms of  0-1 classification loss based on reward margin. The gradient of the reward margin in the multi-token generation scenario is also computed and discussed. In addition, the authors conducted experiments to justify their data  assumptions and theoretical findings.

**Strengths:**

The paper is clear and well-written. The theoretical results on DPO are novel and the assumptions made are justified via experiments.

**Weaknesses:**

* The paper assumes only the last readout layer of the transformer is trainable and the output has the form $y=Wg(x)$, where $g(x)$ is some feature map realized by the previous layers of the transformer. This effectively turns the model into a linear/logistic regression model, although trained on a different loss. Thus the proof techniques follows from standard ODE analysis and I do not find much novelty in the proof.

* The analysis on generalization error is done only for the single-token case. I wonder if similar results can be established for the multi-token case.

**Questions:**

* Is it possible to generalize the analysis in this work to other fine-tuning settings? For example, what if the updates on the weight matrix have low-rank structure as in LORA?

* Line 266-269: what are the values of $x,y$? Is  $y$  a  $|V|$-dimensional vector or a one-dimensional label?

* Typos: Eq (3) misses an expectation on the log-ratio term. Line 233-235: it seems that the conclusions for two the cases are swapped?

---

> ### Author Response · Authors · 2024-11-17
>
> We sincerely appreciate your positive feedback and insightful comments! We address the questions and comments below in detail.
>
> > 1. The paper assumes only the last layer of the transformer is trainable
>
> We appreciate the concerns about this assumption. This choice serves to **strike a good balance between theoretical tractability and practical relevance**. This decomposition allows us to rigorously capture core dynamics driving generalization, and extract valuable insights into the factors that influence preference learning. The reward dynamics involved, even in last-layer, turned out to be quite non-trivial for theoretical analysis. Jointly analyzing all components of the model leads to intractable complexity.
>
> While the analysis is linear with respect to $W$, it's important to note that **the underlying feature representations $g(x)$ still encapsulate the rich non-linear behaviors of the LLM, which is captured by our theoretical analysis**. Specifically, the characterization of non-linear features is detailed in **Section 4.1**, which is empirically validated on real-world LLMs (**Lines 428-464**).  Additionally, this setting is relevant in many parameter-efficient fine-tuning methods, where the feature backbone is often frozen to prevent overfitting, and in black-box scenarios where the backbone is not exposed to the end-user. Therefore, our approach remains relevant to certain practical applications, and is not entirely divorced from practices.
>
>
> We acknowledge that this linearization abstracts away some complexities, such as the interactions between $W$ and the evolving feature backbone. However, our empirical results validate that **this approximation remains effective in capturing the behavior of real-world systems, such as LLaMA-2 fully fine-tuned with preference data, where the feature-map is allowed to change**. As shown in **Lines 466-474**, the results are consistent with our theory, suggesting that the insights gained from our analysis extend to more general settings. Moreover, this decomposition provides a clear and interpretable starting point for understanding reward dynamics and generalization, which can be expanded in future work to include interactions with $g(x)$ or even nonlinear changes in the weight matrix.
>
> For these reasons, we believe our framework provides a foundational step in analyzing preference learning, focusing on the most impactful and mathematically tractable components. We hope this clarification highlights why this decomposition is both a reasonable and productive choice for our theoretical contribution.
>
> > 2. Multi-token case.
>
> That's a great question. Our current analysis focuses on the single-token case as a foundational step, as it provides a more tractable framework to rigorously analyze the generalization error. Extending these results to the multi-token case is an important and challenging open problem.
>
> That said, we believe similar meaningful results can be established for the multi-token case. As outlined in **Section 4.3**, we already analyze the reward dynamics for multi-token scenarios. Specifically, when the reward dynamics for a given example can be shown to lead to a consistently positive or negative reward, we can generalize this result to samples with similar embeddings, ensuring that preferences learned for one example apply reliably to similar cases. This suggests that the insights from our single-token framework are not only foundational but also extendable to capture the dynamics of multi-token generation, which we aim to address more comprehensively in future work.
>
>
> > 3. Other fine-tuning settings
>
>
> Thank you for the insightful question. We believe that the analysis can generalize to other fine-tuning settings, such as those involving low-rank updates like LoRA, as well as to other preference objectives, such as IPO. For example, if we wanted to consider any other objective function $f(r)$ where $r$ is the reward margin, the only part of the original dynamics,
> $$ \tau \dot{r_j} = \frac{1}{N}\sum_{i=1}^N \beta^2 \sigma(-r_i)C(x_i, x_j) $$
> that would change is that $\sigma(-r_i)$ would instead be $-f'(r_i)$. In the case of LoRA, the weight matrix updates are restricted to low-rank subspaces, which could introduce additional constraints on the reward margin dynamics. These constraints could be modeled by incorporating the rank limitation directly into the theoretical analysis. This would introduce non-linearities to the reward dynamics and allow us to study how the low-rank nature of the updates impacts generalization guarantees. We view this as an exciting opportunity for future work and are optimistic that our current analysis can provide a strong foundation for exploring these specialized settings.

---

> > ### Author Response · Authors · 2024-11-17
> >
> > > 4. Lines 266-269
> >
> > $x$ corresponds to an input prompt,  and $y_w$ and $y_l$ are tokens in the vocabulary space $V$ or 1-dimensional labels. Throughout the paper, we use the boldface $y$ for one-hot vectors as mentioned in lines 201-203.
> >
> > > 5. Eq (3) and Lines 233-235
> >
> > Thank you for the correction on Eq (3), which has been fixed! For lines 233-235, we believe the cases described are correct. If all tokens are different, there are no indices that have non-zero elements in both $(y_{w,j}-y_{l,j})^\top (y_{w,i}-y_{l,i})$. However, if the preferences are the same, the dot product will have two indices where the elements multiply to 1, which gives the factor of 2.

---

> > > ### Comment · Reviewer_WjKN · 2024-11-26
> > >
> > > Thanks for the response! My questions are answered. I will keep my score.

---

> > > > ### Author Response · Authors · 2024-11-27
> > > >
> > > > Thank you for the feedback and for supporting our work!

---

### Official Review · Reviewer_zqF8 · 2024-11-01

**Soundness:** 3
**Presentation:** 2
**Contribution:** 2
**Rating:** 5
**Confidence:** 2

**Summary:**

This paper provides a finite time theorectical guarantee for the generalization capability of the preference learning algorithms, with primary focus on Direct Preference Optimization algorithm.

**Strengths:**

1. Novelty of Research Question: Generalization capability of preference optimization algorithm is of crucial importance, either in theory or practice. This paper provides analysis towards theorectically understanding the generalization guarantees of alignment algorithms, defined as averge 0-1 loss in distinguishing prefered from non-preferred answers. Limited theorectical works have been done for understanding the success of these algorithms, and this paper could possibly entrigger further research along this line to improve the understanding of at least offline RLHF algorithms.

2. Clarity of Introduction and notations: The introduction part and preliminaries are clean and well written.

**Weaknesses:**

1. Unclear assumption on LM: On line 187-188, the authors made an important assumption/simplied setting by describing the model outputs as $Wg(x)$. This simplification lacks relevant discussion or introduction, and is hard for readers like me to understand the intuitions of this assumption and following arguments. In addition, since nowadays (almost) all LLM models are autoregressive, this model assumption in the paper can not incoporate the stochasticity, which seems to be way too strong. How to understand that the deterministic models weights lead to random outputs in this setup (otherwise how preference pairs are formulated)?

2. Overall, I think that the section 3.2 is not well written and very hard to follow, especially several key assumptions or results are missing or not clarified. Equation (6) should be a continuous approximation to the SGD of the loss function in (4) from my understanding, but no derivation or illustration is given. For a theorectical oriented paper, this section needs additional heavy polishing for a clearer formulation of the model class to be analyzed.

3. Theorem 4.2's bound looks to be quite vacuous. I did a simple check of math, looks like the bound is meaningless until Q is extremely large (which is impossible if you need that many samples for preference learning from a simple example of k clusters). The bound seems to thus have very limited practical meanings.

**Questions:**

Is there any connection of paper to the analysis of generalization error of PbRL in the literature? See other questions in weakness.

---

> ### Author Response · Authors · 2024-11-17
>
> We sincerely appreciate your feedback and insightful comments! We are encouraged that you recognize the importance and novelty of our research focus. We address the questions and comments below in detail.
>
> > 1. Unclear assumption on LM, concern on the stochasticity
>
> We would like to clarify that **our abstraction of the model output does not alter the autoregressive nature of LLMs nor prevent stochasticity in learning**. We are happy to explain this in detail below.
>
>
> **First, modeling $Wg(x)$ is fully compatible with the autoregressive nature of LLMs and is consistent with both single-token prediction and multi-token predictions (Section 4.3).** Here $g(x)$ encodes the non-linear, autoregressive features extracted by the LLM, and $W$ maps these features to the output logits (with the same dimensionality as the number of tokens). For example, in multi-token predictions, as described in Section 4.3 (**Lines 340-367**), the autoregressive nature is preserved by conditioning the prediction of each token $y_t$ on the preceding tokens $y_1, \dots, y_{t-1}$:
>   $$
>    p(y_1, \dots, y_T) = \prod_{t=1}^T p(y_T | y_1, \dots, y_{t-1}),
>    $$
>    where $p(y_t | y_1, \dots, y_{t-1})$ is based on $\text{softmax}(Wg(x))$, with $g(x)$ corresponding to the embedding at the current step, and the tokens lying in the vocabulary space $\mathcal{V}$. This abstraction aligns fully with the operation of LLMs in practice.
>
> **Second, $W$ is updated stochastically, allowing adaptation to the preference data distribution.** By treating $W$ as a trainable component optimized via stochastic gradient descent, our abstraction captures the inherent stochasticity in learning while focusing on key factors influencing preference learning. This allows us to isolate the dynamics of preference learning while maintaining analytical tractability. Importantly, the responses in the dataset are fixed, and we update the output distribution directly with respect to these fixed preferred/not-preferred responses, rather than modeling a sampled generation.
>
> Lastly, it's important to note that **the underlying feature representations $g(x)$ still encapsulate the rich non-linear behaviors of the LLM, which is captured by our theoretical analysis**. Specifically, the characterization of non-linear features is detailed in **Section 4.1**, which is empirically validated on real-world LLMs (**Lines 428-464**).  Additionally, this setting is relevant in many parameter-efficient fine-tuning methods, where the feature backbone is often frozen to prevent overfitting, and in black-box scenarios where the backbone is not exposed to the end-user. Therefore, our approach remains relevant to certain practical applications, and is not entirely divorced from practices.
>
> Our empirical results further validate that our theory **remains effective in capturing the behavior of real-world systems, such as LLaMA-2 fully fine-tuned with preference data, where the feature map is allowed to change**. As shown in **Lines 466-474**, the results are consistent with our theory, suggesting that the insights gained from our analysis extend to more general settings. Moreover, this decomposition provides a clear and interpretable starting point for understanding reward dynamics and generalization, which can be expanded in future work.
>
> For these reasons, we believe our framework provides a foundational step in analyzing preference learning, focusing on the most impactful and mathematically tractable components. **In the revised manuscript, we will expand on these points and ensure the connection between our abstraction and the autoregressive nature of LLMs is explicitly clarified**. Thank you for your thoughtful feedback.
>
>
> > 2. Presentation of Section 3.2
>
> Thank you for the careful read and helpful feedback. While we dedicated significant effort to polishing our paper, we agree that Section 3.2 could benefit from additional work. We will provide a more detailed illustration of Equation (6) as a continuous approximation to the SGD in Equation (4). As suggested, we will carefully revise this section to better articulate the underlying assumptions, conditions, and the model class being analyzed, and ensure the presentation is precise and accessible. Your comments are greatly appreciated!

---

> > ### Author Response · Authors · 2024-11-17
> >
> > > 3. Theorem 4.2's bound looks to be vacuous
> >
> > You raise an insightful point. Theorem 4.2 is primarily intended to illustrate the dependence on key factors such as the number of clusters and the sample size $Q$. This loosening of the bound is intentional to highlight the relationships between these variables, providing theoretical insight into how they influence generalization in preference learning.
> >
> > To address concerns about practicality, **we provide a tighter bound in the Appendix**, where it is shown that with smaller variance in the preference data, the bound becomes non-vacuous. Furthermore, this bound could be further tightened by adjusting certain assumptions, such as introducing a low-rank covariance structure for the clusters, which would more realistically model many real-world scenarios. We will revise the discussion of Theorem 4.2 in the main text to make these points clearer, ensuring the reader understands the trade-offs between simplicity and tightness of the bounds.
> >
> > > 4. Is there any connection of paper to the analysis of generalization error of PbRL in the literature?
> >
> > Thank you for raising this point. While there may be conceptual links to the analysis of generalization error in PbRL, the settings we consider are quite distinct. PbRL typically focuses on on-policy reinforcement learning, whereas our work examines offline direct optimization of the policy.
> >
> > We provide a theoretical framework that connects reward margin dynamics to generalization bounds, which, to the best of our knowledge, has not been explored in PbRL literature. However, we acknowledge that there may be connections worth exploring further, and we appreciate your suggestion. In our revisions, we will add a more detailed discussion of how our work complements or contrasts with generalization analyses in PbRL literature, addressing potential overlaps and distinctions explicitly.

---

> ### Comment · Reviewer_zqF8 · 2024-11-24
>
> I would like to thank the authors for their detailed responses. Despite this, my concerns are not addressed.
>
> > assumptions on LM in Sc3.2
>
> I am still puzzled about $Wg(x)$. For a theorectically-oritended paper, a clear explanation of theorectical assumptions and why these assumptions should be of crucial importance, and again Sc3.2 is hard to follow with many elements not explained. Can you update Sc 3.2 with more explanations on what are the dimensions, what can this assumption represent?
>
> Although the authors do admit that Equation (6) are some continuous approximations, but necessary details are still not provided.
>
> > Theorem 4.2's bound looks to be vacuous
>
> For the updated bound, I am also still confused so far. When the $\epsilon$ is small, the exponential term is close to 1, this bound is bounded by like $K\cdot Q^2$, which grows quadractically with the sample size $Q$. I do not fully understand why this provides a sharper bound, and could provide meaningful pratical guidance.
>
> ---
>
> Overall, given the current manuscript (which authors haven't updated the main contents yet as far as I can see), although I like the interesting topic and this research direction the paper is focusing on, I think it's hard for readers who may come from practitioners of alignment fields to understand the whole logic of this paper and to seek relevant insights provided.
>
> In the rebuttal, the authors have mentioned the updates to be done in the revised manuscript. I have been waiting to see it, but haven't seen that yet, for which I would like to kindly remind that ICLR allows you to update your manuscript towards a better version. Since the current manuscript still does not address my concerns, I will keep my score.

---

> > ### Author Response · Authors · 2024-11-26
> >
> > We appreciate the feedback and insights! We have updated the manuscript with the revisions and have provided clarification on the setting in section 3.1 as well as the explanation on gradient flow. For Theorem 4.2, the bound improves as $\epsilon$ grows and this occurs when the variance decreases, so when the variance is small, the bound can be non-vacuous. Again, this bound could be further tightened by adjusting certain assumptions, such as introducing a low-rank covariance structure for the clusters, which would more easily allow for non-vacuous bounds in real-world scenarios.

---

### Official Review · Reviewer_GhgS · 2024-11-04

**Soundness:** 2
**Presentation:** 2
**Contribution:** 2
**Rating:** 5
**Confidence:** 4

**Summary:**

This paper provides the first theoretical analysis of generalization in preference learning, specifically for Direct Preference Optimization (DPO). Through a novel theoretical framework, the authors analyze how well models trained with DPO can generalize to unseen data by examining the reward margin and its trajectory during training. The paper's key contribution is deriving learning guarantees showing that DPO-trained models can correctly discern preferred responses on unseen data with high probability under specific conditions. The analysis reveals that generalization capability depends on factors like the number of preference concepts in the dataset and the similarity between different responses. The theoretical insights are validated through empirical experiments on contemporary large language models.

**Strengths:**

The paper presents several notable strengths. First, it offers a rigorous theoretical foundation for understanding generalization in preference learning - an area that previously lacked thorough theoretical analysis despite its practical importance. Second, the paper successfully bridges theory and practice by providing concrete bounds and conditions under which DPO can generalize, with these insights validated through careful empirical studies. Third, the analysis of reward margin dynamics provides actionable insights into how different factors (like number of concepts and sample size) affect generalization. Finally, the work demonstrates strong technical depth while maintaining practical relevance, as shown through experiments on both synthetic data and real-world language models.

**Weaknesses:**

**Major Concerns:**

1. The theoretical analysis decomposes the model output into the product of W g(x), whereas we only consider the dynamic of W. It is reduced to a linear analysis.

2. The assumption of mixture of Gaussians distribution of the distribution of preference is rather interesting. However, it is verified the cosine similarity and orthogonality between persona. It is still unclear why we could have a Gaussian distribution for each cluster.

Therefore, the two main bounds look fine, but these too points make the theoretical analysis not very strong.

**Clarity Issue:**

The paper has inconsistent descriptions about what component performs the prediction:

1.	Line 70: "The loss can correctly predict all training samples into the preferred vs. non-preferred categories"

2.	Line 280: "the implicit reward model from DPO can correctly predict all training samples into the preferred vs. non-preferred categories"

First, "implicit reward model" is not defined. From lines 159-162 it seems to be the same thing as reward margin, but this should be clarified.

Second, both "loss" and "implicit reward model" are weird subjects here. Actually, we use only the LLMs rather than the implicit reward model to predict. I sense the authors don't use LLM $\pi_\theta$ as the subject and say "The trained LLM can correctly predict all training samples into the preferred vs. non-preferred categories" because this statement would be incorrect.
This makes the analysis confusing - the generalization guarantee applies to the implicit reward model, but what we actually use is the LLM. Actually, there could not exist such a guarantee for $\pi_\theta$, creating a disconnect between theory and practice.


**Minor Concern:**

1. In the claim

"While existing generalization theory often focuses on overparameterized models achieving near-optimal loss or models independent of the training process, our framework rigorously assesses how well models generalize after a finite number of gradient steps, reflecting real-world LLM training practices"

and the discussion of related work, the authors seem ignore another line of generalization bound research called algorithmic stability or uniform stability, which also bound the generalization gap by training steps T.

2. Llama 2-7B is relatively old, experiments on 3/3.1/3.2 would be better.

**Questions:**

see weakness

---

> ### Author Response · Authors · 2024-11-17
>
> We sincerely appreciate your constructive feedback and insightful comments! We are encouraged that you recognize our contribution of offering rigorous theoretical foundation for preference learning and bridging theory and practice. We address the questions and comments below in detail.
>
> > 1. The theoretical analysis decomposes the model output into the product of W g(x), whereas we only consider the dynamic of W. It is reduced to a linear analysis.
>
> We appreciate the concerns about this assumption. This choice serves to **strike a good balance between theoretical tractability and practical relevance**. This decomposition allows us to rigorously capture core dynamics driving generalization, and extract valuable insights into the factors that influence preference learning. The reward dynamics involved, even in the last-layer, turned out to be quite non-trivial for theoretical analysis. Jointly analyzing all components of the model leads to intractable complexity.
>
> While the analysis is linear with respect to $W$, it's important to note that **the underlying feature representations $g(x)$ still encapsulate the rich non-linear behaviors of the LLM, which is captured by our theoretical analysis**. Specifically, the characterization of non-linear features is detailed in **Section 4.1**, which is empirically validated on real-world LLMs (**Lines 428-464**).  Additionally, this setting is relevant in many parameter-efficient fine-tuning methods, where the feature backbone is often frozen to prevent overfitting, and in black-box scenarios where the backbone is not exposed to the end-user. Therefore, our approach remains relevant to certain practical applications and is not entirely divorced from practices.
>
>
> Lastly, our empirical results validate that **this modeling remains effective in capturing the behavior of real-world systems, such as LLaMA-2 fully fine-tuned with preference data, where the feature map is allowed to change**. As shown in **Lines 466-474**, the results are consistent with our theory, suggesting that the insights gained from our analysis extend to more general settings. Moreover, this decomposition provides a clear and interpretable starting point for understanding reward dynamics and generalization, which can be expanded in future work to include interactions with $g(x)$ or even nonlinear changes in the weight matrix.
>
> For these reasons, we believe our framework provides a foundational step in analyzing preference learning, focusing on the most impactful and mathematically tractable components. We hope this clarification highlights why this decomposition is both a reasonable and productive choice for our theoretical contribution.
>
> > 2. Clarification on the Gaussian distribution
>
> Thank you for raising this important question. The assumption is motivated by the fact that Gaussian distributions are sufficient to capture the variance of embeddings while enabling a more straightforward and tractable analysis. This choice allows us to focus on deriving meaningful generalization bounds and reward dynamics without introducing unnecessary complexity.
>
> **To support this assumption, we have conducted empirical verification based on the Llama-2 model**. Specifically, we observe that across different personas, the average norm of the covariance matrix is 0.058, and the average squared distance from the mean is 0.227. These empirical findings suggest that the Gaussian assumption provides a reasonable approximation of the underlying structure of preference data. While the covariance structure could be further refined, we opted for a single parameter $v$ to enhance interpretability.
>
> We also recognize that real-world preference distributions might deviate from strict Gaussianity. Future work could explore more flexible models that better capture the nuances of preference data, such as non-parametric clustering methods or other distributional assumptions. For this work, the Gaussian mixture model serves as a practical and reasonable starting point, enabling us to make progress on the challenging problem of understanding generalization in preference learning.

---

> > ### Author Response · Authors · 2024-11-17
> >
> > > 3. Clarity Issues
> >
> > Thank you for the careful read and catching these clarity issues!
> >
> > **Implicit reward model**. According to the DPO loss, the implicit reward for the preferred response is
> > $$
> > r(x_i, y_{w,i}) = \log \frac{\pi_\theta(y_{w,i|x_i})}{\pi_\text{ref}(y_{w,i|x_i})},
> > $$
> > where $\pi_\theta$ is the current model and $\pi_\text{ref}$ is the base model; and similarly the implicit reward for the less preferred response is $\log \frac{\pi_\theta(y_{l,i|x_i})}{\pi_\text{ref}(y_{l,i|x_i})}$.
> >
> > Hence, **DPO can be interpreted as optimizing an implicit reward model** by trying to maximize the difference between the implicit reward of the preferred response and the non-preferred response, i.e., $$
> > \log \left(\frac{\pi_\theta(y_{w,i|x_i})}{\pi_\text{ref}(y_{w,i|x_i})} - \frac{\pi_\theta(y_{l,i|x_i})}{\pi_\text{ref}(y_{l,i|x_i})}\right).
> > $$
> >
> > In other words, it optimizes for a positive reward margin (with additional scaling of $\beta$). For a given sample, a positive reward margin means that the current LLM distinguishes the preferences for that sample better than the original model. Having an implicit reward model that correctly predicts all training samples into preferred vs nonpreferred categories means that the LLM has been updated in a way that more closely follows the preference data for all examples. Essentially our generalization guarantee is saying that the updated LLM will consistently be more likely to output the preferred responses.
> >
> > We will update the paper to more clearly explain the implicit reward model.
> >
> > > 4. Works on algorithmic stability
> >
> > Thank you for bringing up these works and we will update the paper to mention this line of work and update the claim accordingly!
> >
> > > 5. Experiments on more recent model
> >
> > We completely agree! Admittedly, our research was conducted prior to the release of LLaMA 3.1. We have additionally verified our theoretical findings on the latest version, and they remain consistent. We will update the manuscript to reflect these new results.

---

### Official Review · Reviewer_6LVT · 2024-11-06

**Soundness:** 3
**Presentation:** 3
**Contribution:** 2
**Rating:** 5
**Confidence:** 3

**Summary:**

This paper attempts to understand the generalization of models trained using DPO. Different from classic generalization theory, which often assumes the underlying model is overparameterized and with near-zero loss (i.e., can perfectly fit the training data), the assumption in the paper is more realistic and tailored to LLMs. The authors considers scenarios where LLMs are fine-tuned over a few gradient steps rather than extensive training. The key idea is to analyze each sample's reward margin and track how it evolves during training. The authors derived the conditions to keep the reward margin positive for all training samples.. Additionally, they also provide generalization  bounds for new samples from the same preference distribution. The theoretical results were also validated empirical.

**Strengths:**

1. The theoretical results derived in this paper provide a better understanding on the dynamics of DPO training, which are helpful for the community to understand the mechanism of preference training.

2. The empirical verification shows the theory aligns with the practice to some extent. This confirms the usefulness of the theories, though the empirical verifications are still very limited.

**Weaknesses:**

1. Typically, in RLHF, the choice of algorithm doesn't matter much; most of the effort goes into working on RLHF infrastructure and data. DPO is a promising technique to reduce infrastructure demands, but it still shows a gap compared to on-policy algorithms like PPO—assuming the infrastructure is well-optimized. This may limit the real-world impact of follow-up works, but I think it's a relatively minor issue here.

2. I highly doubt that ignoring the fact that the LLM is pre-trained would lead to any meaningful theoretical results. This consideration is crucial in explaining why we need much less post-training data to align LLMs. For small or under-trained LLMs, this result may not hold. Therefore, if the derived result is agnostic to this setting, I am not convinced it will be useful or even correct. However, I understand the challenge in mathematically formulate a "pretrained  LLM".

3. For LLMs, I think the traditional generalization bound analysis doesn't seem particularly interesting (i.e., in-distribution). The post-training data is relatively small and cannot cover all the cases of user queries, yet the LLM can still generalize to most cases that have not seen anything similar in the training dataset. To work on something really impactful and solid, I think it's essential to have a careful and thorough thinking about the above mentioned factors.

**Questions:**

See weakness.

---

> ### Author Response · Authors · 2024-11-17
>
> We sincerely appreciate your feedback and insightful comments! We address the questions and comments below in detail.
>
> > 1. Clarification on pre-trained LLM
>
> We fully agree with your viewpoint on considering the fact that LLM is pre-trained. **We indeed considered pre-trained LLM in our theoretical analysis, making our results practically relevant**. In particular, we characterize our data distribution based on pre-trained LLMs and how DPO optimizes with respect to this data distribution. This characterization is detailed in **Section 4.1**, which is empirically validated on real-world, pre-trained LLMs, such as LLaMA-2, and alignment datasets designed to capture the challenges posed by diverse human preferences (**Lines 428-464**). Specifically, we characterize the structure of preference data, comprising of a shared component and concept-specific directions, to derive stronger guarantees compared to scenarios with less structured preference data. Our empirical validation demonstrates that our theory applies to pre-trained models undergoing fine-tuning for alignment. We do observe the empirical trends align with the generalization bounds, which validates the correctness of our theory.
>
> It is worth noting that pretraining enables data efficiency by structuring embeddings in ways that facilitate subsequent fine-tuning. Our results align with this intuition, as demonstrated in our generalization guarantees. Pretraining contributes to these factors by embedding high-quality priors, implicitly enhancing alignment performance with less preference data. While our framework does not directly model the pretraining process, the theoretical results are consistent with and supported by the benefits of pretraining.
>
> Finally, we acknowledge that certain aspects of pretraining, such as its specific inductive biases, could further refine our framework. However, this does not diminish the utility or relevance of our analysis. Instead, it underscores the generality and extensibility of our theoretical framework. For instance, future work could incorporate additional assumptions about pretrained representations, such as their cluster structure or alignment with preference datasets, to derive more specialized results. In this work, our focus has been on building a theoretical foundation that can be applied broadly, allowing capturing the dynamics of alignment for both well-pretrained and less-pretrained models.
>
> > 2. For LLMs, I think the traditional generalization bound analysis doesn't seem particularly interesting (i.e., in-distribution)
>
> You raise an insightful point. This is something we have also thought about deeply.
>
> Although LLMs exhibit impressive generalization capabilities, enabling them to handle many cases they have not explicitly seen, they also frequently fail in situations where the data is similar to what they have encountered. For preference learning, this inconsistency underscores the importance of understanding when and why LLMs behave as expected.
>
>
> A critical first step is determining whether the model can reliably distinguish what is preferred from what is not. This is the central question we address in our work, and it is foundational to ensuring alignment in real-world applications. For example, even in settings that should theoretically be straightforward, such as in-distribution preference prediction for HH-RLHF-trained models, test accuracy can be only 75%. **This highlights that even within the same distribution, generalization is far from perfect or well-understood**. Scaling to more complex preference distributions makes generalization even harder, as our theoretical framework indicates. These observations reinforce the importance of focusing on this classical setting and ensuring it is thoroughly understood before extending the analysis to more complex out-of-distribution scenarios.
>
> Lastly, our work provides a deeper understanding of multi-token generation by analyzing reward dynamics, which further contributes to unraveling why and when LLMs align with expected behaviors. Future work could leverage our theoretical framework as a foundation to address the broader and impactful question of OOD generalization in preference learning. We recognize that they represent expansive and complex problems that cannot be addressed comprehensively in the scope of a single work. This makes it all the more important to take measured steps, starting with a thorough understanding of in-distribution generalization as a foundation.
>
>
> > 3. Gap between DPO and on-policy: This may limit the real-world impact of follow-up works, but I think it's a relatively minor issue here.
>
> Thank you for pointing this out, and we will keep this in mind!

---

### Author Response · Authors · 2024-11-17

**Review summary**

We sincerely appreciate all four reviewers for their time and effort in providing feedback and suggestions on our work. We are encouraged that reviewers recognize the significance, novelty, and practical relevance of our work, as highlighted by their comments:

- _"The theoretical results derived in this paper provide a **rigorous understanding** on the dynamics of DPO training - an area that previously lacked thorough theoretical analysis despite its practical importance_" (R1, R2)
- _"the paper successfully **bridges theory and practice** by providing concrete bounds and conditions under which DPO can generalize, with these insights validated through **careful empirical studies**....the work demonstrates **strong technical depth** while maintaining **practical relevance**"_ (R2)
- _"Novelty of Research Question: Generalization capability of preference optimization algorithm is of crucial importance...this paper could possibly **entrigger further research along this line** to improve the understanding of at least offline RLHF algorithms."_ (R3)
- _"The paper is **clear and well-written**. The theoretical results on DPO are **novel** and the **assumptions made are justified** via experiments."_ (R4)



**Responses and changes in the manuscript**

In addition, we greatly appreciate the constructive feedback from the reviewers, which further strengthens our work. Many of the concerns raised are already addressed in the current version of the paper, and we provide additional clarifications and explanations to all questions raised. Here is a summary of the major changes we will make following the valuable suggestions from the reviewers:

- Provide further clarification on key assumptions and expand discussions to ensure these points are more explicit and accessible to readers.
- Add a definition of the implicit reward model and its connection to the model output.
- Revise Section 3.2, including a more detailed derivation and illustration of Equation (6) as a continuous approximation to the SGD.
- Add clarification on Theorem 4.2 and its tighter version in the Appendix, emphasizing its practical relevance.
- Expand the discussion of related works, including algorithmic stability and PbRL.

**Broader Impact**
As commended by multiple reviewers, our work addresses a critical gap in understanding the generalization dynamics of preference optimization algorithms, laying a rigorous foundation for future research in aligning LLMs with human preferences. The strong theoretical grounding combined with empirical validation demonstrates the soundness and practicality of our approach. We believe our work will trigger a line of future works to deepen the understanding.

Below, we address each reviewer’s comments point by point.

---

> ### Author Response · Authors · 2024-11-24
>
> Dear reviewers,
>
> We understand it’s been an overwhelming season, and everyone is juggling many responsibilities. This makes us especially grateful for the time and attention you’ve given to our submission despite these challenges.
>
> We hope our responses have addressed your concerns and clarified the key points of our paper. As a theory-focused work, we have made every effort to carefully validate our assumptions and theoretical results based on real-world LLMs. Closing the gap between theory and practice is a non-trivial moonshot, and our work represents a significant stride toward narrowing it. We believe this contribution will inspire a line of future works to deepen the theoretical understanding of alignment, which is urgently needed for the community.
>
> If there are any remaining questions or suggestions, we welcome your feedback and will address them promptly. Thank you again for your thoughtful insights and for supporting this important area of research.
>
> Sincerely,
>
> Authors

---

### Meta-Review · Area_Chair_5QP6 · 2024-12-21

**Metareview:**

This paper provides a theoretical framework for understanding generalization in preference learning using DPO. The key contributions include an analysis of reward margin dynamics, finite-step gradient-based training guarantees, and empirical validation on LLMs, such as LLaMA-2. The paper aims to bridge theory and practice, addressing a critical gap in understanding preference learning for aligning LLMs with human preferences.

**Strengths:**
* The paper addresses an important problem in preference learning, offering a rigorous theoretical perspective.
* Novel analysis of reward margin dynamics provides insights into generalization behavior.
* Empirical validation supports the practical relevance of the theoretical framework.

**Weaknesses:**
* The reliance on simplified assumptions, such as training only the last layer, limits the generality of the results.
* Clarity issues, especially in Section 3.2, make the exposition difficult to follow.
* Practical significance of the derived bounds, such as Theorem 4.2, is limited without stricter conditions.
* Limited consideration of pretraining dynamics affects applicability to modern LLMs.

In summary, the paper provides interesting insights but falls short of the bar for acceptance due to reliance on simplified assumptions, limited theoretical contributions, and unresolved clarity issues. Future iterations should expand the scope of the analysis, address pretraining dynamics, and improve technical clarity to maximize impact.

**Additional Comments On Reviewer Discussion:**

During the post-rebuttal discussion, reviewers highlighted that the reliance on simplified assumptions, such as training only the last layer, limits the generality and theoretical significance of the results. Concerns were also raised about the practical relevance of the derived bounds (e.g., Theorem 4.2) and insufficient clarity in key sections, particularly Section 3.2. While the empirical validation is appreciated, the theoretical framework's limited applicability and lack of consideration for pretraining dynamics remain key weaknesses.

---

### Decision · Program_Chairs · 2025-01-22

Reject